# PHOREAU v1.0: a new process-based model to predict forest functioning, from tree ecophysiology to forest dynamics and biogeography

- Tanguy Postic<sup>1</sup>, François de Coligny<sup>2</sup>, Isabelle Chuine<sup>1</sup>, Louis Devresse<sup>1</sup>, Daniel Berveiller<sup>3</sup>, Hervé Cochard<sup>4</sup>, Matthias Cuntz<sup>5</sup>, Nicolas Delpierre<sup>3</sup>, Emilie Joetzjer<sup>5</sup>, Jean-Marc Limousin<sup>1</sup>, Jean-Marc Ourcival<sup>1</sup>, François Pimont<sup>6</sup>, Julien Ruffault<sup>6</sup>, Guillaume Simioni<sup>6</sup>, Nicolas K. Martin-StPaul<sup>6\*</sup>, Xavier Morin<sup>1\*</sup> (\*co-last authors)
- 10 ¹CEFE, CNRS, Univ. Montpellier, EPHE, IRD, Montpellier, France.
  ²AMAP UMR931, Botany and Computational Plant Architecture, INRAE, TA A-51/PS2, Boulevard de la Lironde, 34398 Montpellier Cedex 5, France.
  - <sup>3</sup>Université Paris-Saclay, CNRS, AgroParisTech, Ecologie Société Evolution, 91190, Gif-sur-Yvette, France. <sup>4</sup>INRA, UMR 547 PIAF, F-63100 Clermont-Ferrand, France.
- <sup>5</sup>Université de Lorrain, AgroParisTech, INRAE, UMR Silva, 54000 Nancy, France.
   <sup>6</sup>INRAE, URFM, Domaine Saint Paul, INRAE Centre de recherche PACA, 228 route de l'Aérodrome, CS 40509, Domaine Saint-Paul, Site Agroparc, France.

Correspondence to: Tanguy Postic (tanguy.postic@cefe.cnrs.fr)






Abstract. Climate change impacts forest functioning and dynamics, but large uncertainties remain regarding the interactions between species composition, demographic processes and environmental drivers. While the effects of changing climates on individual plant processes are well studied, few tools dynamically integrate them, which precludes accurate projections and recommendations for long-term sustainable forest management. Forest gap models present a balance between complexity and generality and are widely used in predictive forest ecology, but their lack of explicit representation of some of the processes most sensitive to climate changes, like plant phenology and water use, puts into question the relevance of their predictions. Therefore, integrating trait- and process-based representations of climate-sensitive processes is key to improving predictions of forest dynamics under climate change.

In this study, we describe the PHOREAU model, a new semi-empirical forest dynamic model resulting from the coupling of a gap model (FORCEEPS), with two process-based models: a phenology-based species distribution model (PHENOFIT) and a plant hydraulics model (SurEAU), each parametrized for the main European species. The performance of the resulting PHOREAU model was then evaluated over many processes, metrics and time-scales, from the ecophysiology of individuals to the biogeography of species.

PHOREAU reliably predicted fine hydraulic processes at both the forest and stand scale for a variety of species and forest types. This, alongside an improved capacity to predict stand leaf areas from inventories, resulted in better annual growth compared to ForCEEPS, and a strong ability to predict potential community compositions.

By integrating recent advancements in plant hydraulic, phenology, and competition for light and water into a dynamic, individual-based framework, the PHOREAU model, developed on the Capsis platform, can be used to

understand complex emergent properties and trade-offs linked to diversity-effects effects under extreme climatic events, with implications for sustainable forest management strategies.

1 Introduction

Forests cover approximately 30% of the Earth's land surface, hosting the majority of terrestrial biodiversity. They are crucial carbon sinks (Pan et al., 2011), play a vital role in climate regulation (Chapin III et al., 2008), and provide essential ecosystem services to humans (Nadrowski et al., 2010). However, climate change poses significant risks to forests, including disruptions to forest dynamics (McDowell et al., 2020a), as increasingly extreme environmental conditions have profound effects on forest structure and composition as well as on forest functioning, including massive mortality events (Allen et al., 2010). Such impacts are assessed through experimental (Decarsin et al., 2024; Gavinet et al., 2019a) and empirical (McDowell et al., 2020a) studies. Yet, although such approaches are key to understanding and anticipating forests' response to climate change, they cannot cover the entire spectrum of environmental contexts, species compositions, and forest history. By filling those gaps in knowledge, forest models represent key complementary tools to effectively investigate the combined impacts of species composition and climate change on forest dynamics and functioning (Bugmann, 2001; Maréchaux et al., 2021).

Yet the robustness of such models — most often calibrated on historical data — is often questioned when used to make predictions for the uncertain transition period of the coming decades (Parmesan, Morecroft and Trisurat, 2022; Van der Meersch et al., 2025). Focusing on Europe, climate projections generally describe drier conditions, with might lead to a shift from light to soil water as the main limiting resource over which individual trees compete (McDowell et al., 2020a). In this context, the accuracy of forest projections might depend in large part on whether models are able to account for causal relationships between water stress and stand composition (Brodribb et al., 2020; McDowell et al., 2022; Van der Meersch et al., 2025). For example, instead of postulating general *a priori* species complementarity effects in resource use, process-based modelling must strive to capture how individual trees harness and compete for light and water *in natura*.

Furthermore, depicting and understanding the role of diversity in ecosystem functioning has been a key focus of ecological studies for at least two decades (Hooper et al., 2012; Kinzig et al., 2002; van der Plas, 2019). In forest ecosystems, the importance of the role of diversity — both structural and compositional — on productivity and wood biomass has been firmly established by numerous studies over a wide range of conditions and methods (Liang et al., 2016; Morin, 2011; Nadrowski et al., 2010; Paquette and Messier, 2011; Ratcliffe et al., 2017). In addition, there is some evidence that tree diversity could modulate the resistance and recovery of forest productivity under stress or disturbance (Ammer, 2019; Blondeel et al., 2024; Jourdan et al., 2019; Schnabel et al., 2021), although the level of consensus varies with the type of stress or disturbance considered (Decarsin et al., 2024; Messier et al., 2022). Yet despite these patterns, there remains a scarcity of data regarding the actual differences in functioning of monospecific and mixed forests, and their relative response to changing climate conditions. In fact, while the diversity-productivity relationship is well evidenced — a global meta-analysis has shown mixed-species stands were on average 25% more productive than their respective species' monocultures (Zhang, Chen and Reich, 2012) —, data regarding the link between species diversity and the ability to withstand

extreme climatic events is more scarce and contradictory. Where some studies have linked forest diversity to a lessened sensitivity of tree growth to drought (Lebourgeois et al., 2013; Anderegg et al., 2018; Serrano-León et al., 2024), others have found this relationship to be strongly context-dependent (Grossiord et al., 2014; Forrester et al., 2016; Jactel et al., 2017), and restricted to dry environments. Moreover, with the rapid shift in climatic conditions, it would be a mistake to assume that the same patterns of diversity-productivity and diversity-resilience relationships used to support the stress-gradient hypothesis (Bertness and Callaway, 1994) will apply in the next decades to newly drought-prone sites, where water resource limitation has not had the chance to shape the coevolution of the local species over the past millennia. In fact, the same structural and specific complementarities that are currently responsible for increasing the productivity of existing mixed temperate forests through a better usage of the light resource could become a source of vulnerability, as competition for water intensifies proportionally to the density and foliage areas of the stands (Decarsin et al., 2024; Haberstroh and Werner, 2022; Jucker et al., 2014; Moreno et al., 2024).

For these reasons, and because experimenting composition effects in mature forests is especially difficult, the evaluation of diversity effects in forest ecosystems has also increasingly relied on forest models, particularly gap models based on processes (Bohn and Huth, 2017; Jonard et al., 2020; Maréchaux and Chave, 2017; Morin et al., 2021). Indeed, the prospective power of such models make them key tools in testing various hypotheses on the diversity-functioning link (De Cáceres et al., 2023a; Maréchaux et al., 2021), but also in evaluating forest management practices that incorporate species mixing (Jourdan et al., 2021) and more generally in simulating forest-response to the long-term impacts of climate change (Reyer, 2015).

To improve our ability to forecast the impact of climate change on forests and to better test adaptation solutions related to composition and management, we have thus identified, through a review of the available literature and consistently with recent studies (Bugmann and Seidl, 2022), two main weaknesses common in forest gap models: their modelling of regeneration (Price et al., 2001), and of tree mortality (Keane et al., 2001). These shortcomings can respectively be traced to a lack of explicit representation of phenological processes on the one hand, and hydraulic processes on the other.

In fact, there is a lack of knowledge regarding the effects of species mixing on forest resistance and resilience to drought, although trait-data describing the hydraulic functioning of tree species has been steadily accumulating in the last years. A great variety of water-stress adaptation and drought response strategies among species have been identified (Choat et al., 2018; Martin-StPaul et al., 2017): these include traits linked to the allocation between transpiring and conducting surfaces, stomatal control and conductance (Johnson et al., 2012), water storage, root-to-shoot ratio, specific leaf area, safety margins (Martin-StPaul, Delzon and Cochard, 2017), and rooting depths (del Castillo et al., 2016). These traits and their variability ultimately account for many of the plant-to-plant interactions responsible for water-competition reduction and facilitation (De Cáceres et al., 2021; Moreno-de-Las-Heras et al., 2023; Moreno et al., 2024; Mas et al., 2024). However, understanding their net impact in existing forests is complicated by environmental and structural variability among stands, and more generally by the fact that the most common available indicators — growth and mortality — integrate over time many processes that are difficult to unravel. Therefore, although the dynamic and integrative effect of species-mixing on medium-term

drought-resilience most directly concerns forest management strategies elaborated today, it is difficult to formulate *a priori* recommendations. Decoupling the effects of hydraulic trait diversity from forest structure (foliage area, tree density) involves significant methodological difficulties (Forrester and Pretzsch, 2015), and is further complicated by the feedbacks between traits and stand structure (Guillemot and Martin-StPaul, 2024), as trees have been shown to adapt hydraulic to the forest structure (Limousin et al., 2012, 2022; Martin-StPaul et al., 2013; Moreno et al., 2024).

Furthermore, even disregarding species diversity, the relationship between forest structure, density and productivity is itself poorly understood: there is no consensus on the link between tree-size heterogeneity and productivity (Bourdier et al., 2016; Dănescu et al., 2016; Pretzsch and Biber, 2010), and while stand density has been statically correlated with increased growth (Forrester, 2014; Reineke, 1933), it is the overall dynamic interactions between these factors that must be understood (Morin et al., 2025). The prohibitive cost of testing all the factors affecting forest functioning (species diversity, stand structure and density, response to climate and soil conditions, effect of management...) in experimental or observational studies further justifies the use of forest ecosystem models (Pretzsch et al., 2017), which are able to replicate *in silico* the complex plant-to-plant interactions that regulate competition for above- and belowground resources, evaluate potential facilitation and competition reduction processes, and integrate them over time in stand structure dynamics that account for trade-offs between drought-resistance and productivity.

Recent gap models (Maréchaux and Chave, 2017; Morin et al., 2021) by explicitly modelling crown sizes and species shade tolerances, have focused on capturing the processes through which canopy packing and spatial niche partitioning can emerge. However, space is not only the dimension through which plant species partition resources – time is also an important vector of asymmetry through which different species can coexist in by exploiting different niches (Gotelli and Graves, 1996). Relative shifts of even a few days in leaf phenology – either through earlier budding or later senescence – have been shown to have major impacts on plant growth, by allowing otherwise shaded understory plants to receive full sunlight (Jolly, Nemani and Running, 2004). As warming climate conditions advances the phenology of most species, increasing productivity (Park et al., 2016) at the expense of additional vulnerability to spring frosts (Lopez et al., 2008), accurately integrating phenological responses of individual species is an important next step in improving the ability of gap models to represent competition for light.

In addition, phenological processes (including seed production, leaf dormancy and resistance to frost) have been shown to be major factors in determining species distribution (Chuine, 2010). Indeed, while many studies highlight the role of species diversity in forest functioning, it is important not to lose sight of the fact that the presence of a species in a given forest is itself the result of a complex historical process conditioned both by site conditions and species coexistence mechanisms. By directly integrating trait-based phenology, gap models can therefore more accurately capture this dynamic by making species diversity an emerging factor of the modelling framework.

Process-based forest models like gap models — originally inspired by the pioneering JABOWA model (Botkin et al., 1972) — are part of a broader class of vegetation demography models (VDMs) that explicitly simulate the

birth, growth, and death of trees within forest stands (Bugmann and Seidl, 2022; Fisher et al., 2018; Scheiter et al., 2013). These models aim to bridge physiological processes at the plant scale with ecosystem and land surface dynamics, offering a detailed, mechanistic representation of vegetation dynamics under changing environmental conditions. VDMs have been increasingly used in Earth system modelling because they allow for the exploration of how shifts in species composition, resource competition, and trait diversity influence ecosystem resilience and carbon balance (Fisher et al., 2015). However, demographic processes such as tree mortality and regeneration — long recognized as critical drivers of ecosystem resilience, biogeochemistry, and post-disturbance recovery (Seidl and Turner, 2022) — remain underrepresented in many models whose development has primarily been focused on growth (Bugmann and Seidl, 2022). As climate change leads to no-analog environmental conditions (Williams and Jackson, 2007) the robustness of model projections will depend largely on how well these demographic responses are captured, particularly in relation to competition, drought, and disturbance.

Here we introduce PHOREAU, a new process-based forest gap model that extends the capabilities of classical models by incorporating key demographic processes with fine-grained physiological realism. Developed within Capsis (Dufour-Kowalski et al., 2012), a modular software platform designed to simulate the growth and management of forest stands, PHOREAU integrates and builds upon three well-established models — ForCEEPS, PHENOFIT, and SurEau — to simulate forest dynamics under changing climate conditions. In particular, the model extends the scope of classic gap models by including a detailed representation of plant water use and competition for the water resource as well as a detailed representation of plant phenology and its impact on reproduction and frost leaf damage. The PHOREAU model thus presents a coupling between recent advances in the process-based modelling of plant water relations under conditions of extreme drought (Cochard et al., 2021a; Ruffault et al., 2022) with state-of-the art phenology (Chuine and Beaubien, 2001) and light competition (Morin et al., 2021) models, in an individual-based gap-model capable of simulating most types of forest structures (Morin et al., 2025) and forest management (Jourdan et al., 2021). The validity of this approach is underpinned by its reliance on species-specific hydraulic, allometric and phenological traits, grounded in decades of experimental research (Cochard et al., 2021a; Kattge et al., 2020; Leinonen, 1996).

The PHOREAU model has been designed to shed light on some of the many pending issues regarding the effects of species diversity on forest functioning, such as the impact of extreme droughts (Piedallu et al., 2023) or the role of complementarity in leaf phenology on growth in mixed stands (Morin, 2011). More generally, the model offers the opportunity to tackle issues ranging from the physiology of individuals to the biogeography of species. Therefore, our multi-stage validation protocol, presented here, involves daily hydraulic processes, yearly productivity, pluri-annual mortality, and long-term species composition.

## 2 Presentation of the model

The PHOREAU model builds on three process-based models, which have been presented in previous publications. For the sake of clarity, we have chosen to summarize only the main processes of each model, and to focus on the integration methodology and the new processes allowed by the coupling. These notably include detailed representations of competition for light and water, with multi-layered representations of both tree canopies (Fig. W1) and rooting systems (Fig. W2) disaggregated at respectively daily and hourly time-steps, with the resulting aggregated shade and drought stress factors of individual trees being integrated in the yearly regeneration, growth and mortality equations presented in the following section. Refer to Fig. 1 for a schematic representation of the PHOREAU model, to Fig. 3 for a more detailed breakdown of the coupling between the ForCEEPS, PHENOFIT and SurEau models which constitute PHOREAU, and to table Z1 for a summary of the main processes' time steps and models of origin.

**Figure 1** | Schematic representation of the PHOREAU model. The principle of the three main demographic processes (growth, mortality, regeneration) and competition for light are inherited from the ForCEEPS forest dynamics model. Tree hydraulics and competition for water and tree foliar phenology come from the coupling with the SurEau and PHENOFIT models, respectively.

## 2.1 The forest community gap-model

#### 2.1.1 Core demography equations

245

250

In PHOREAU, forest dynamic processes (growth, mortality and recruitment) retain the overall structure of those at the core of the ForCEEPS model (Morin et al., 2021). ForCEEPS (Forest Community Ecology and Ecosystem Processes) is a gap model that relies on a few ecological assumptions to simulate the dynamics of tree establishment, growth and mortality in independent small patches of land, that are aggregated to derive properties at the forest scale. While the model is not spatially explicit at the patch level, it is individual-based: two trees of the same species and the same age can have different growth rates under the same climate, depending on the specific patch-level biotic constraints of light-competition. Derived from the FORCLIM model (Bugmann, 1996; Didion et al., 2009) the ForCEEPS model was developed with the aim of simulating forest dynamics under a wide range of environmental conditions while limiting the need for prior calibration, and was designed to be equally able to simulate planted, managed, or natural forests (Morin et al., 2020, 2025).

Tree growth is computed at a yearly time-step in two phases. First maximum diameter increment is calculated using an empirical equation shown in Eq. 1, as a function of trunk diameter at breast height at the start of the year, and a maximum species growth rate  $g_s$ .  $b_s$  and  $c_s$  are species specific allometric parameters (respectively derived from  $H_{max}$  and s), and  $H_{max}$  the maximum height reachable by that species. Height is directly linked to diameter following another species-specific allometric parameter.

235 
$$\Delta D_{opt} = g_s \times \frac{D \times \left(1 - \frac{H}{H_{max,s}}\right)}{2 \cdot H_{max,s} - b_s \times e^{(c_s \cdot D) \times (c_s \cdot D + 2)}}$$
 Eq. 1

Then, realized growth is determined from optimal growth after reduction by a series of growth-reduction factors (bounded between 0 and 1) following a modified geometric mean, as shown in Eq. 2.

$$\Delta D = \Delta D_{opt} \times \sqrt[3]{GR_{light} \times GR_{gdd} \times GR_{drought} \times GR_{soil}} \times GR_{crown}$$
 Eq. 2

Drought, growing degree days, and soil reduction factors are determined by site soil and climatic conditions, and modulated by species-specific parameters. The other factors represent biotic constraints related to light availability.  $GR_{light}$  represents the immediate effect of competition for light, and depends on the cumulated leaf area above or at the same level as the considered tree.  $GR_{Crown}$  represents the long-term effects of crown size reduction on the capacity of trees to grow and assimilate carbon. With the exception of  $GR_{Crown}$ , the reductors are calculated by comparing fixed species response parameters to yearly aggregated biotic or abiotic factors.  $GR_{soil}$  is unchanged from ForCEEPS, with a set site richness parameter constant throughout the simulation. While the formulas for  $GR_{gdd}$  and  $GR_{light}$  are also unchanged, the underlying growing degree-days (GDD) and light availability (LA) values are calculated at a much finer grain taking advantage of newly integrated leaf phenology (see Sect. 2.4.3) and stand microclimate (see Appendix F). Likewise,  $GR_{drought}$  is now the result of a detailed representation of

stand hydraulics presented in Sections 2.2 and 2.4.2. Finally,  $GR_{crown}$  remains the ratio of realized to potential tree crown size; but trees in PHOREAU can see their crown reduced through drought-stress and frost damage components, on top of the light-suppression mechanism already implemented in ForCEEPS, as shown in Eq. 3. This is a first approach, following Wang, Zhou and Wang, (2021). We are aware this representation is incomplete, and does not account for leaf regrowth, or differential effects according to tree age and size: the absence of an explicit representation of source and sink compartments, and the lack of tree age data to implement an age-differentiated response to leaf loss, was a limiting factor. Refer to Eq. 25 and Eq. 26 in Section 2.4.3 for definitions of the new frostComponent and droughtComponent.

$$GR_{crown} = Min(lightComponent \times frostComponent \times droughtComponent, 1)$$
 Eq. 3

Similarly, tree establishment is regulated by winter temperature, growing degree days, light availability, and stand browsing intensity. First, a yearly number of potential seedlings  $n_{PotentialSeedlings,s}$  for a given species is determined, by multiplying a species shade tolerance parameter kLa (shade intolerant species having a greater regeneration potential) with a new reproductive success factor  $R_s$ , which is calculated at a yearly time-step for each species as the product of the proportion of uninjured flowers and the proportion of fruits that reach maturity (see Sect. 2.3 for a presentation of the underlying phenology model). Once the number of potential seedlings for a given species has been determined (Eq. 4), the probability of establishment of each individual seedling  $P_{est,s}$  is formally unchanged from the ForCEEPS framework (Eq. 5, with details in Morin et al., 2021), but indirectly integrate the refinements presented in below in the calculation of phenology and microclimate (through  $P_{GDD}$ ), light availability at soil level (through  $P_{LA}$ ), and soil water balance (through  $P_{DT}$ ). Finally, each selected sapling is initialized with a DBH of 1.27 cm.

$$n_{PotentialSeedlings,s} = 0.006 \times patchSize_{m^2} \times kLa \times R_s$$
 Eq. 4

$$P_{est,s} = P_{T_w} + P_{GDD} + P_{Dr} + P_{Br} + P_{LA} + c_{est}$$
 Eq. 5

Tree mortality is the combination of a stochastic background process combining stand density and tree longevity, and a growth-related mortality that represents stress-caused tree death linked to biotic and abiotic constraints. In addition, PHOREAU mortality also integrates a new cavitation mortality mechanism ( $P_{cavitationMortality}$ ) described in Section 2.4.2 and Eq. 20. With  $P_0$  and  $P_g$  respectively the background and growth-related mortality components described in Morin et al., (2021), the chance that a given tree dies on a given year is such that:

$$P_{mort} = P_{cavitationMortality} + \left(1 - P_{cavitationMortality}\right) \times max(P_0, P_g)$$
 Eq. 6

A full description of the ForCEEPS model developed on the Capsis modeling platform (Dufour-Kowalski et al., 2012) that was used as a base for this study can be found in Morin et al., (2021). In the following section, we present new developments to the representation of canopy structure and light competition that have been included in the ForCEEPS model, before the coupling with SurEau and PHENOFIT.

#### 2.1.2 Improvements to canopy structure and light-competition






In anticipation of the coupling with SurEau and PHENOFIT, a number of modifications were made to the ForCEEPS model, focusing on microclimate, light-dependent height plasticity, and improvements to the light-competition module. This proved necessary when integrating transpiration-driven water fluxes, as stand leaf area is one the main driver of embolism in the SurEau model (Cochard et al., 2021b), and preliminary results indicated a poor capability of the ForCEEPS to reproduce observed leaf area indices from stand inventory, in both relative and absolute terms. These refinements are summarized below and in Fig. 2, with more in-depth descriptions in supplementary information.

Light-dependent height plasticity: ForCEEPS infers tree height from trunk diameter using fixed allometric relationships, limiting its ability to capture site effects and competition-driven height-diameter variations. In reality, understory trees allocate more growth to height, while trees in low-density stands prioritize diameter growth (Oliver and Larson, 1996), especially in shade-intolerant species (Delagrange et al., 2004). Recognizing this, we have incorporated dynamic height growth in PHOREAU, by adjusting height increments based on competition-driven parameters and species shade tolerance parameter. Refer to appendix A for further details.

Crown-length reversion: The PHOREAU model changes the representation of crown length dynamics by allowing an increase of the crown ratio when tree light availability increases, unlike the ForCEEPS model, which only permitted decline. This modification thus aims to account for the beneficial impact of the death or removal of a tree on neighboring trees, which find themselves with greater access to light than before, and can therefore regrow the lower parts of their crown previously self-pruned due to light competition. This yearly crown ratio increase for previously suppressed trees is capped at 5% of the difference between the previous year's crown ratio, and the potential crown ratio based on light conditions. Refer to appendix B for further details.

- 315 Species-dependent crown shapes: The PHOREAU model improves crown-shape representation by allowing for a greater range of crown shapes than the default ForCEEPS inverse-cone, including ellipsoidal and conical shapes. This in turn allows for a better representation of inter-specific competition, with complementarities arising from differences in crown structure. Refer to appendix C for further details.
- Density-dependent light availability: PHOREAU maintains ForCEEPS' balance between predictive power and computational efficiency by simplifying light dispersion calculations, using a vertical stratification approach without explicit tree positioning. However, this method reduces light competition to a single leaf area index (LAI) value, overlooking horizontal canopy structure and gaps that influence tree growth. To address this, PHOREAU integrates a clumping factor (Ω) into its light extinction coefficient, capturing variations in foliage aggregation and improving realism (Nilson, 1971; Black et al., 1991; Bréda, Soudan and Bergonzini). This approach reflects observed trends, such as the inverse relationship between LAI and light extinction (Dufrêne and Bréda, 1995), and aligns with methods used in remote sensing (Chen et al., 2012; Demarez et al., 2008; Zhu et al., 2018). Refer to appendix D for further details.

Incorporation of Specific Leaf Area (SLA): ForCEEPS crown size allometric relationships, originally calibrated for a few temperate European species (Bugmann, 1996; Burger, 1951), led to inaccurate predictions when applied to a broader range of species, particularly Mediterranean and understory trees. PHOREAU addresses this by recalculating tree foliage area using species-specific leaf area (SLA) values, improving the model's ability to represent interspecific differences in drought resistance, in addition to other traits described in table S13, as tree water use is driven by leaf area. Refer to appendix E for further details.

Microclimate derived from stand-structure: Forest canopies buffer climatic conditions in the understory, resulting in cooler, more stable daytime temperatures and warmer nighttime temperatures compared to the canopy. This microclimate effect is especially pronounced in dense, structurally complex canopies (De Frenne et al., 2021), helping young understory trees resist drought despite shallow root systems (Forrester and Bauhus, 2016). Because the PHOREAU model integrates fine-scale hydraulic and phenological mechanisms within a forest-structure gap model, it is able to capture these effects of microclimate on plant functioning. In particular, we integrate microclimatic temperatures and vapor-pressure deficits derived from macroclimate data using a statistical model based on a slope and equilibrium approach presented in Gril et al., (2023) and Gril, Laslier, et al., (2023), incorporating patch characteristics like leaf area index (LAI), maximum tree height, and vertical complexity index (VCI). Hourly microclimate temperatures are then used to calculate vapor pressure deficits for transpiration computations, as well as degree-day accumulation for tree growth and regeneration. Refer to appendix F for further details.



Crown-length Bootstrapping. To avoid initial oscillations in stand leaf area resulting from year-wise adjustments of tree crown length based on above leaf area, an algorithm, presented in Appendix M, was developed to initialize all tree crown lengths at equilibrium values at the beginning of the simulation. This allows modeled forest inventories to immediately start simulations with realistic foliage areas, which, in ForCEEPS and earlier models, would have taken several years of iterations to achieve.

**Figure 2** | Presentation of the modifications in the light-competition module between ForCEEPS (Morin et al., 2021) and PHOREAU, with a description of the main changes

## 2.2 SurEau: a plant hydraulics model

The SurEau model (Cochard et al., 2021; Ruffault et al., 2022) is a model of the SPA family (soil-plant-atmosphere, Mencuccini et al., 2019), dedicated to model plant response during extreme drought, which describes water flows in a soil, plant and atmosphere system. It was developed with the idea (1) simulating the water status of plants throughout a complete drying sequence going beyond stomatal closure, including plant desiccation and hydraulic failure (Choat et al., 2018); and (2) of being able to be initialized from accessible environmental data (climate, description of the structure of the forest stand by inventory or remote sensing) and hydraulic "traits" at fine taxonomic grains (species, provenance, etc.) which are increasingly available in global databases (e.g. Martin-StPaul, Delzon and Cochard, 2017; Guillemot et al., 2022). The SurEau model uses daily meteorological data as inputs, which are then disaggregated into hourly values; among its outputs are the time to full stomatal closure, and the hourly level of cavitation of each organ. There are two published versions of SurEau and their detailed presentation can be found in Cochard et al., (2021) and Ruffault et al., (2022). These two versions differ in the complexity of the hydraulic architecture of the plant and the numerical scheme used to solve the equations of transport (Ruffault et al., 2022).

We describe below in a synthetic manner the main principles of the model, the equations used for the coupling, and its implementation in Phoreau. For the purpose of the coupling, we have recently implemented a highly modular version of SurEau into the Capsis platform using Java object-oriented programming, which includes the main aspects of both previous versions of SurEau. The specific functioning of each compartment is elegantly implemented using object-oriented principles, allowing for modularity and clarity in the model design.

SurEau includes principles of forest water balance such as transpiration, rainfall interception, soil evaporation, rain infiltration into different soil layers, and water drainage into deep reservoirs. The specificity of SurEau is to explicitly represent water transport within the tree through a system of resistance and capacitance (Fig. 3). This hydraulic architecture makes it possible to calculate the water status (water potential and water content) at different levels of the tree and the soil. The tree's organs (e.g., roots, trunk, branches, leaves) are represented by a water compartment separated into a symplasm and an apoplasm. The symplasm corresponds to the water reservoir made up of living tissues (parenchyma, phloem, etc.); it is elastic and can exchange water with the vascular system under the effect of tissue volume variations. The apoplasm, in contrast, consists of non-living tissues such as xylem vessels and cell walls, forming a rigid, low-capacitance pathway that facilitates bulk water transport but stores little water.

The soil-plant-atmosphere system is modeled through different compartments ("hydraulic cells"), considered as "computational entities" and implemented as classes in Java, which are interconnected and exchange water fluxes through specific functions which model ecophysiological processes. This Capsis version builds on the implementation of generic computational entities that we call *SPH* (Soil-Plant-Hydraulic) compartments, which can be attributed a specific type (soil, symplasm, apoplasm). Each type is defined by specific functions to compute water potential and water quantities. These *SPH* compartments can be connected together to build a tree of any possible complexity. The fluxes between cells are determined with Fick's law by using the water potential gradients between cells and their hydraulic conductances. The water content of each cell is therefore described as

the result of inflows and outflows; and the water potential of each cell is calculated with the appropriate formulation according to the nature of these cells (soil, symplasm, apoplasm). For the soil a water retention curve is used (Van Genuchten, 1980). For the symplasm, the law of pressure-volume curves (Tyree and Hammel, 1972), which expresses the relationship between water content and water potential, is used to describe loading and unloading dynamics. These laws can be parameterized using abundant pressure-volume curve data (Bartlett et al., 2012). The effect of cavitation is to alter the hydraulic conductance of the apoplasm, and can lead to hydraulic failure. However, cavitation also releases apoplastic water into the transpiration stream, which can temporarily attenuate the drop in water potential (i.e., water stress). Both phenomena are irreversible (but see Sect. 2.4.2). The percentage loss of conductance (PLC) through vessel embolism is calculated using the water potential of the organ's apoplasm ( $\psi_{Apo}$ ) and an empirical sigmoid function described by species-specific inflexion and slope parameters ( $P_{50}$ , slope $P_{cav}$ ) as shown in Eq. 7:

$$PLC = \frac{100}{1 + \left(e^{\frac{slope_{cav}}{25} \times \left(\psi_{Apo} - P_{50}\right)}\right)}$$
 Eq.7

PLC is a key indicator of the risk of mortality by hydraulic failure, and has been elected a key variable for the coupling with ForCEEPS (see Sect. 2.4.2).

The main fluxes from the plant to the atmosphere are the stomatal and the cuticular transpirations. Cuticular and stomatal transpirations are computed using gas-phase conductance, and the vapor pressure deficit between the organ and the atmosphere. The leaf stomatal and cuticular conductance are connected in parallel to produce the leaf conductance, itself connected in series to other boundary and crown conductances to produce the overall canopy conductance. Leaf cuticular conductance varies with leaf temperature and photosynthetic activity. Meanwhile, stomatal conductance is calculated as the product of a maximum stomatal conductance without water stress  $g_{stom,clim\_max}$  (which ranges between species specific parameters  $g_{stom\_max}$  and  $g_{stom\_night}$  depending on light, temperature, and CO2 concentration), with a regulation factor  $\gamma$  based on plant water status, as shown in Eq. 8.

$$g_{stom} = g_{stom,clim\_max} \times \gamma$$
 Eq. 8

In particular  $\gamma$  represents the degree of stomatal closure between 0 and 1, computed using leaf symplasm water potential  $\psi_{LSym}$  and a sigmoid function described by inflexion and shape parameters  $\psi_{gs50}$  and  $slope_{gs}$  as shown in Eq. 9 (these parameters are themselves derived from species-specific pressure-volume curve parameters  $P_{gs12}$  and  $P_{gs88}$ : refer to Ruffault et al., (2022), for more details).

$$\gamma = 1 - \frac{1}{1 + e^{\frac{slope_{gs}}{25} \times (\psi_{LSym} - \psi_{gs50})}}$$
 Eq. 9

Numerical resolution of the plant water balance is based either on the explicit or the faster semi-implicit method presented in Ruffault et al., 2022. This first version of PHOREAU v1.0 uses the same simplified tree hydraulic architecture as in Ruffault et al., (2022) and uses the faster and generic semi-implicit solver, which solves the water balance equations by assuming that certain variables — cell water potential and stomatal or cuticular transpiration fluxes — stay constant during each small time step; this has the effect of reducing numerical instabilities and increasing runtime by a factor of 10 000. Before performing the coupling, we verified this new implementation could provide nearly identical results as the previous version under the same initial conditions.

#### 2.3 PHENOFIT: a phenology-based distribution model

The PHENOFIT model (Chuine & Beaubien 2001) is a process-based species distribution model for temperate trees which calculates the probability of presence over several years of a given species for a particular set of environmental conditions. This probability is derived from the estimated fitness of an average adult individual of that species, which is itself the product of the probability to survive until the next reproductive season, and the probability to produce viable seeds by the end of the annual cycle. The model assumes that survival and reproduction depend on the synchronization of tree development to seasonal climatic variations, with the plasticity of key phenological events such as leaf unfolding, flowering, fruit maturation, and leaf senescence. The model uses soil data and daily meteorological data (minimum and maximum temperature, rainfall, relative humidity, global radiation, and wind speed) as inputs. It is composed of several sub-models: phenological models for leafing, flowering, fruiting and leaf senescence (for reviews refer to Chuine and Régnière, 2017, and Chuine et al., 2024); a frost injury model (Leinonen, 1996); a survival model; and a reproductive success model calculated as the proportion of uninjured fruits that reach maturation considering photosynthetic ability and the proportion of leaves not killed by frost (Chuine and Beaubien, 2001). A visual representation of the model can be found in Fig. 3.

In PHENOFIT, both the leafing and the flowering dates  $(t_f)$  are calculated with a two-phase phenology model. In the first phase of endodormancy (Eq. 10), the bud must be exposed to a certain amount  $(C_c)$  of chilling units  $(R_{c,t})$  from the onset of dormancy  $(t_0)$  in order to break this endodormancy at date  $t_1$ . In the second phase of ecodormancy, or quiescence (Eq. 12), the bud cells elongate in response to forcing temperatures. They must accumulate forcing units  $(R_{f,t})$  until a threshold value  $(F_c)$  is reached, that corresponds to the leafing or flowering date. The type of response functions to temperature are identical for leafing and flowering, only the parameters of these functions differ between the two. Calculations are done at daily time-step, using mean daily temperatures  $(T_t)$  and species-specific parameters (a, b, c, d, e) as shown in Eq. 11 and Eq. 13. Leaf senescence dates  $t_c$  are calculated following the model of Delpierre et al., (2009).

Flowering and leafing dates are then used, alongside the daily minimum temperature  $(T_i)$  between bud onset and leaf senescence or fruit maturation, to determine proportions of leaves and flower-fruits  $(I_l, I_f)$  uninjured by frost. The probability that fruits reach maturation  $(I_r)$  is calculated on the basis of the proportion of uninjured leaves which produce the assimilates accumulated in the fruits, the date of flowering from which thermal energy can begin to be accumulated  $(t_{flowering})$ , and a species-specific parameter  $E_c$  representing the average amount of energy needed to reach maturation (Eq. 11). Finally, a yearly probability of producing viable seeds, or reproductive

success (R), is calculated as the product of the probability that fruits will ripen and the proportion of uninjured fruits reaching maturation, as shown in Eq. 17.

480

$$C_{c} = \sum_{t_{0}}^{t_{1}} R_{c,t}$$
 Eq. 10
$$R_{c,t} = \frac{1}{1 + e^{c(T_{t} - e)^{2} + d(T_{t} - e)}}$$
 Eq. 11
$$F_{c} = \sum_{t_{1}}^{t_{f}} R_{f,t}$$
 Eq. 12
$$R_{f,t} = \frac{1}{1 + e^{a(T_{t} - b)}}$$
 Eq. 13
$$I_{l} = f(t_{leafing}, T_{l})$$
 Eq. 14
$$I_{f} = f(t_{flowering}, T_{l})$$
 Eq. 15
$$I_{r} = f(t_{flowering}, I_{l}, E_{c})$$
 Eq. 16

$$I_l = f(t_{leafing}, T_i)$$
 Eq. 14

$$I_f = f(t_{flowering}, T_i) Eq. 15$$

$$I_r = f(t_{flowering}, I_l, E_c)$$
 Eq. 16

$$R = I_f I_r$$
 Eq. 17

For each organ and each species, parameters are inferred statistically using time series of phenological observations from native populations (dates of leaf unfolding, senescence, flowering, and fruit maturation) for different sites and different years, or from experimental results found in the literature (resistance of plant organs to frost).

As the model simulates one average individual, it does not take into account demography or biotic interactions with other species. It also does not represent the impacts of plant growth on survival and resource allocation, but takes into account the effect of a reduction of leaf area on survival. While it can (by calibrating parameters from phenological data of different provenances) represent the way phenological plasticity can vary from one site to another due to genetic differentiation and eventual local adaptation, we have chosen here to use only one calibration set per species: in other words, we account for the plastic response of a species to varying climate conditions, but not for the genetic differentiation of this response. As a result, species performance may be underestimated at the limits of its distribution due to non-representative parameter estimates.

510




Our main concern when coupling the PHENOFIT into PHOREAU was avoiding that some processes shared by the models be taken into account more than once. For example, we could not directly use the global plant fitness output of PHENOFIT, nor its plant survival output, which integrates drought-effects already represented by the SurEau model. In the end, we used four main yearly PHENOFIT outputs: leaf unfolding and senescence dates  $(t_f, t_c)$ , the percentage of uninjured leaves not damaged by frost  $(I_l)$  and reproductive success (R).

# 2.4 PHOREAU: the coupled model

#### 2.4.1 Model-coupling framework









At the heart of the PHOREAU model is the integration of the ForCEEPS, SurEau and PHENOFIT models. This integration was made possible by the presence of all three models on the Capsis Java platform (Dufour-Kowalski et al., 2012). The Capsis simulation platform has been continuously developed since 1994, hosting many models pertaining to various aspects of forest dynamics. Its generic and flexible architecture allows modelers to integrate various aspects of forest dynamics, while its interactive simulation mode facilitates applications in teaching and decision support for forest stakeholders.

Two major considerations guided the coupling of the models: avoiding overlapping processes, and minimizing the increase in computing time that might arise when integrating models operating at different time-scales. In its simplest state, the connection between the three models can be described as follows. Independent PHENOFIT simulations are first run for each species and climate year, whose outputs (dates of leaf unfolding and senescence, probability of reproduction) are then read and fed into the main PHOREAU simulation.

At the beginning of each PHOREAU simulation year, all the trees currently present in the plot are used to initialize a separate SurEau simulation. This simulation lasts exactly one year, using the same daily climate as the main simulation, albeit with a further hourly disaggregation required by the Sureau numerical scheme. In addition to species hydraulic traits parameters (see Ruffault et al., 2022), morphological (i.e. size dependent) variables (including tree volume computed from height and diameter, as well as leaf area, PLC, and light availability), are retrieved directly from the main ForCEEPS simulation; leafing and senescence dates are obtained from PHENOFIT; and the initial state of the soil is retrieved from its state at the end of the previous SurEau simulation for year n-1. Throughout the simulation data is collected and sent back to the main ForCEEPS simulation to determine the effects of drought stress on growth, mortality, and defoliation, as detailed in the following sections.

However, the sub-hourly time-scales of the SurEau processes, which represent a roughly tenfold increase in computation time, warranted the implementation of two major optional simplifications to this framework. They are summarized below, with more in-depth descriptions in supplementary information (Appendices G and H)

Treewise aggregation for SurEau module. SurEau simulation runtimes are primarily influenced by the number of distinct SPH-compartments, and particularly the number of trees. To optimize runtime, PHOREAU reduces the number of trees simulated by SurEau each year, while maintaining the overall stem volumes and foliage areas at the stand, species and cohort level. This is achieved through an aggregation method that groups trees into a predefined number of classes per species (set to 3 in our model evaluation), preserving structural integrity while simplifying competition for water by reducing the number of trees. Trees are distributed into a configurable number of classes based on trunk diameter, separating for example mature and juvenile trees. As trees grow, they

may shift between classes, and some classes may remain empty in certain years. Each class is represented by a single aggregate tree, whose characteristics are determined by summing (volume, foliage area, biomass) or averaging (height, root depth, light availability) the corresponding attributes of the individual trees. At the end of each year, aggregated class results are uniformly distributed among the trees that make them up, informing yearly growth and mortality equations (trees of a given class suffer the same growth reduction due to stress, and have the same probability of dying due to cavitation). This method significantly reduces computational complexity, while maintaining key ecological dynamics in SurEau.

**Figure 3** | Detailed representation of the processes included in the SurEau, ForCEEPS and PHENOFIT models. Red circles indicate outputs used for the coupling, and red lines their destination in the ForCEEPS simulation.

## 2.4.2 Drought-stress integration




PHOREAU accounts for drought impacts on tree growth and mortality thanks to the integration of the SurEau plant hydraulics model. Drought-induced mortality can occur either directly — in response to extreme drought through high level of xylem embolism leading to hydraulic failure — or as a long-term consequence of reduced growth related to consecutive low intensity drought and defoliation. As a result, the model effectively represents the interplay between the short term extreme drought effect of hydraulic failure, and the longer term drought effect carbon starvation (McDowell et al., 2008).

Drought feedback on growth in PHOREAU is assessed by using the factor of stomatal aperture  $\gamma$  computed by SurEau at the tree level. This replaces the ForCEEPS formulation, where a growth reduction factor  $GR_{drought}$  was

computed by comparing a drought index (DrI) based on a simple monthly water budget with an empirical speciesspecific drought tolerance index (Bugmann and Solomon, 2000). The factor of stomatal aperture  $\gamma$  is computed (Eq. 18) from the leaf water potential on the basis of a sigmoid curve described by two species-specific traits ( $P_{gs12}$ the water potential causing 12% stomatal closure, and  $P_{gs88}$  the water potential causing 88% stomatal closure, Cochard et al., 2021b, Ruffault et al., 2022). Daily stomatal apertures are then integrated annually, over the vegetation period, to compute the DrI (Eq. 19). Refer to Appendix I for more details.


580

$$\gamma = 1 - \left(1 + e^{\frac{P_{L,sym} - 0.5 \times (P_{gs12} + P_{gs88})}{0.25 \times (P_{gs12} - P_{gs88})}}\right)^{-1}$$

$$Eq. 18$$

$$DrI = 1 - \frac{1}{n} * \sum_{j=1}^{n} (\gamma_j)$$

$$eq. 19$$

$$DrI = 1 - \frac{1}{n} * \sum_{j=1}^{n} (\gamma_j)$$
 Eq. 19

n: days in year; j: day of year

Drought feedback on mortality and defoliation. Two additional drought stress mechanisms derived from the level of embolism were implemented in PHOREAU. First, drought-induced defoliation was computed on a daily basis for each tree by using the percentage of the leaf xylem embolism (Cakpo et al., 2024). The defoliation rate was set proportional to the embolism rate, with a minimal threshold set at 10% (Eq. 26). The resulting defoliation percentage is applied to the maximum leaf area of the tree for the given day (itself the result of the species crown allometry, reduction of crown length due to competition for light, and the phenological stage of the leaf derived from PHENOFIT) to obtain the effective daily leaf areas used throughout the model, from plant water usage to light competition and rain interception (refer to Sect. 2.4.3 for details and equations). Furthermore, an average yearly defoliation percentage is computed for integration in the  $GR_{crown}$  growth-reductor, which represents the impact of leaf-loss on carbon assimilation and tree growth reduction (Eq. 3), and which, in PHOREAU, is computed as the result of leaf-loss induced by light-suppression, frost, and drought (see Eq. 25 to 27). Finally, the longer-term adaptation between water stress and reduced leaf area is partially captured by the fact cavitation is carried over from year to year, with a specific repair mechanism described below. Refer to Appendix J for more details.

Second the rate of embolism (assessed through the percent loss of cavitation, PLC) is used to estimate extreme drought induced mortality. The PLC computed by SurEau is retrieved for each tree at the end of the year. Because no cavitation-repair mechanism is implemented at this intra-yearly timescale, the end-of-year value is also necessarily the maximal reached PLC. Then, the resulting PLC<sub>%</sub> is converted into a probability of death, which is applied at the end of the year like the other death probabilities in the model (Eq. 6). When the tree aggregation option (see Appendix G) is used, each individual tree of a class receives the drought-induced death probability of its corresponding aggregate tree, and death events are drawn independently among them. The actual conversion of the level of cavitation into a death-probability follows a logistic distribution fitted using data from Hammond

et al. (2019). The probability distribution is parametrized using a constant steepness parameter, and a species-

specific  $LD_{50}$  parameter which corresponds of a point of no return, the lethal dose of cavitation at which exactly 50% of individuals of the species are expected to die (see Eq. 20). As a first approach this  $LD_{50}$  was fixed parameterized at respectively 50% and 80% for gymnosperm and angiosperm species (Choat et al., 2012); Delzon and Cochard, 2014), reflecting the capacity of the latter species to operate at water potentials below the  $P_{50}$  line. This is a result of differences in strategies between embolism-tolerant and embolism-avoidant species, as gymnosperms tend to operate at wider safety margins with vessels more resistant to embolism (Choat et al., 2012). Finally, an additional threshold parameter was added to avoid spurious mortality events for low PLC values, considering even well-watered trees show some degree of embolism throughout the year (Cruiziat, Cochard and Amiglio, 2002). Refer to Appendix K for more details.

$$P_{cavitationMortality} = \begin{cases} (1 + e^{-\lambda * (PLC_{\%} - LD_{50,s})})^{-1} & PLC_{\%} > PLC\_threshold \\ 0 & PLC_{\%} \le PLC\_threshold \end{cases}$$
 Eq. 20

s: species; PLC<sub>%</sub>: end-of-year loss of conductance percentage; LD<sub>50,s</sub>: species cavitation sensibility parameter;  $\lambda$ : steepness parameter (default 0.12); PLC\_threshold: default 20%

Year-to-year cavitation memory and repair. The impact of cavitation on tree functioning can continue long after the end of the initial drought event, and is one of the main causes for the increased vulnerability to future drought events of previously weakened trees (Anderegg et al., 2013; Feng et al., 2021). On the other hand, internal repair mechanisms linked to plant growth (formation of new vessels) can allow the recovery of initial conductance over time (Brodribb et al., 2010). As such, the recovery from embolism in PHOREAU is driven by basal area growth — or, more precisely, by the relative increase of sapwood area, which contains the living conductive vessels. While all new growth is naturally sapwood, as a tree becomes larger the relative proportion of sapwood to heartwood tends to decreases. It follows that to evaluate the rate of replacement of the conductive vessels, the model must first know the pre-existing area of sapwood. PHOREAU uses the foliage area to determine this quantity, through the application of a species-specific, constant, leaf-to-sapwood ratio, also known as the inverse of the Huber value (Cruiziat et al., 2002). The leaf-to-sapwood ratio is applied to the potential one-sided leaf area of the tree, derived solely from its DBH and allometry parameters, and not its actual leaf area after defoliation through competition, frost or drought. This approach, presented in Eq. 21, assumes the Huber value to be constant: we know that this is in fact an important simplification, and that many species adapt their leaf mass per area to site conditions (Lopez et al., 2008).

$$PLC_{S}^{n+1} = Max(0, PLC_{S}^{n} - 100 * \frac{\Delta BasalArea_{n+1}}{LAp_{n+1} * LA:SA_{S}})$$
 Eq. 21

s: species; n: year; PLC: end-of-year loss of conductance percentage LAp: potential one-sided leaf area; LA:  $SA_s$ : species leaf area to sapwood ratio

2.4.3 Leveraging leaf phenology to temporalize light-competition, growth, and rain interception

Daily competition for light. In ForCEEPS, the way the light availability of each canopy layer is determined by the above total leaf area of the above layers, combined with differentiated shade tolerances between species, allows emergent complementarities in a multi-specific context between shade tolerant and intolerant species, resulting on average in greater total stand leaf area and productivity at the stand level (Morin et al., 2025). But alongside spatial complementarities, there exist temporal complementarities in species usage of light related to different leaf phenology (Gotelli and Graves, 1996).




The PHOREAU model, by integrating leaf phenology simulated by the PHENOFIT model (see Sect. 2.3), accounts for these temporal effects. In particular, the PHENOFIT model calculates two dates based on temperature and photoperiod conditions: the unfolding date  $(t_{f,s,n})$  when 50% of the buds show at least one unfolded leaf (BBCH 15), and the senescence date  $(t_{c,s,n})$  when 50% of the leaves have changed color or have fallen (BBCH 95). This gives us the range of days when each tree bears leaves. In practice, the maximum daily foliage area of a given tree  $(LA_{s,i}^{n,j})$  is derived from its maximum yearly foliage  $LAp_{s,i}^n$  (itself the result of species-specific crown allometry and the light availability of the tree, Eq. 22 and 23), by using the dates of leaf unfolding  $t_{f,s,n}$  and leaf senescence  $t_{c,s,n}$  calculated by PHENOFIT for a given species s for a given year n, as described in Eq. 24.

Using this information required an in-depth reworking of the light-competition module: instead of calculating each layer's light availability at the yearly time-step, daily light availability is now calculated by summing the crown areas of all leaf-bearing trees in the above layers. The final tree light availability is calculated by summing, over all its layers, for all the days for which it is itself bearing leaves, each daily layer light availability. To correct for the fact that tree growth is dependent on heat as well as sunlight, this sum is weighed using daily growing degree days (GDD) values, defined as the difference between the average daily temperature and the  $T_0$  base temperature fixed at 5.5°C. This is a first approach, which heavily weighs summer months where growth may be limited by drought: further developments of the model will take advantage of the coupling with SurEau to incorporate tree drought-stress in the weights. Finally, in addition to being temporalized, this formulation integrates all the refinements to canopy representation described in Sect. 2.1.2.


$$LAp_{s,i}^n = c_2^s \times crownsize_{i,n} \times DBH_{i,n}^{A_2^s}$$
 Eq. 22

$$LAp_{s,i}^{n} = c_{2}^{s} \times crownsize_{i,n} \times DBH_{i,n}^{A_{2}^{s}}$$

$$\begin{cases} c_{2}^{s} = 0.35 * SLA_{s} * 2 & (Deciduous) \\ c_{2}^{s} = 0.45 * SLA_{s} * 2 & (Evergreen) \end{cases}$$

$$crownsize_{i,n} = f(LightAvailability_i)$$
 Eq. 23

$$LA_{s,i}^{n,j} = \begin{cases} 0 & j \leq t_{f,s,n} \\ LAp_{s,i}^{n} * \frac{(j-t_{f,s,n})}{UI_{s}} & t_{f,s,n} < j < (t_{f,s,n} + UI_{s}) \\ LAp_{s,i}^{n} * \frac{(j-t_{c,s,n}-cI_{s})}{cI_{s}} & t_{c,s,n} < j < (t_{c,s,n} + UI_{s}) \end{cases}$$

$$Eq. 24$$

$$j \leq t_{f,s,n} \leq t_{c,s,n} \leq t_{c,s,$$

s: species; i: tree; n: year; j: day of year;  $LAp_{s,i}^n$ : maximum tree yearly leaf area;  $t_{f,s,n}$ : species leaf unfolding date;  $t_{c,s,n}$ : species leaf senescence date;  $UI_s$ : species leaf unfolding interval;  $CI_s$ : species leaf coloration interval;  $I_{l,s}$  species year leaf-loss percentage






Stress-induced defoliation. While ForCEEPS implements a mechanism for competition-driven loss of foliage area, representing the reduction of the crown height of dominated trees as their lower branches die off, it does not incorporate mechanisms of leaf-loss driven by extreme meteorological or hydraulic conditions. Unlike competition-driven branch dieback, leaf-loss caused by extreme weather conditions is not usually accompanied by branch death, does not preferentially target the leaves located in the lower parts of the crown, and can be more quickly reverted with shoot regrowth. These differences justified the implementation in PHOREAU of a new mechanism for transitory leaf-loss, distinct from the reduction of crown size, with no memory from one year to the next. The variables used to drive this leaf-loss are derived from the yearly percentage of frost-damaged leaves  $(I_1)$  and daily leaf cavitation  $(PLC_1)$  values calculated respectively in the PHENOFIT and SurEau models (see Sect. 2.2 and 2.3. The PHENOFIT leaf loss index is calculated using the frost injury model of Leinonen (1996), based on the leaf-phenology, temperature and photoperiod conditions. The SurEau drought-induced leaf-loss is presented in Sect. 2.4.2. The values of frost-induced (frostComponent<sub>s,n</sub>, Eq. 25) and drought-induced leaf loss (droughtComponent<sub>i,s,n</sub>, Eq. 26) are integrated in an overall daily tree-specific defoliation percentage (Eq. 27), allowing the model to reflect strategies of drought acclimation, where defoliation can help some species tolerate drought events (Bréda et al., 2006; Limousin et al., 2022) at the cost of a lowered growth potential. This transitory stress-induced defoliation is combined with the maximum daily foliage area of a given tree  $(LA_{s,i}^{n,J})$  to obtain the effective daily leaf area  $LA_{s,i}^{n,j,effective}$ , as shown in Eq. 28. It is this daily leaf area that is in fine used in all PHOREAU processes, from transpiration, GDD accumulation for growth, to light-competition.

$$frostComponent_{s,n} = 1 - \frac{(1-I_l)}{0.5}$$
 Eq. 25

$$droughtComponent_{i,s,n,j} = \begin{cases} 1 - \frac{1}{(t_c - t_f)} \times PLC_{i,s,n,j} & PLC_{\%} > 10\% \\ 1 & PLC_{\%} \le 10\% \end{cases}$$
 Eq. 26

 $DefoliationPercentage_{i,n,s,j} = frostComponent_{s,n} \times droughtComponent_{i,s,n,j}$  Eq. 27

$$LAp_{s,i}^{n,j,effective} = LA_{s,i}^{n,j} \times \left(1 - \frac{DefoliationPercentage_{i,n,s,j}}{100}\right)$$
 Eq. 28







s: species; i: tree; n: year; j: day of year;  $t_f:$  species leaf unfolding date;  $t_c:$  species leaf senescence date;  $I_{l,s}$  species year leaf-loss percentage

This simplified formulation has the disadvantage of disregarding intra-specific differences in phenology arising from differences in size or competition-status (Augspurger and Bartlett, 2003; Gill et al., 1998; Gressler et al., 2015; Vitasse, 2013). Furthermore, it does not yet take full advantage of the PHOREAU hydraulic submodule to account for the effects of drought on leaf development, either through earlier leaf coloration (Xie et al., 2018) or shifted unfolding (Cleland et al., 2007). Further developments of the PHOREAU model should therefore strive to use information from the light competition and water stress modules to inform the calculation of phenology dates.

Growing-degree-days. Furthermore, in addition to plant fluxes and light-competition, leaf phenology was also used to inform the period during which growing degree days (GDD) are accumulated for deciduous species. Evergreen species are assumed to accumulate energy throughout the year. As the ForCEEPS framework worked at a monthly time-step, it was necessary to update the model to calculate GDD using daily temperature data. This introduces both inter-species variability in growth, but also intra-species variability between sites and years. This change impacts both growth (through the temperature growth-reduction factor  $GR_{gdd}$ ) and probability of establishment ( $P_{GDD}$ ). See Eq. 29 or the updated calculation of annual GDD sums, including phenology and microclimate, of a tree of species s and average weighted foliage height h, with  $T_0$  the base temperature ( $T_0 = 5.5^{\circ}C$ ).

$$GDD_{h}^{s} = \sum_{j=t_{f,s}}^{t_{c,s}} max (T_{h}^{j} - T_{0})$$
 Eq. 29

The rain interception module. In addition, PHOREAU integrates a rain interception module that reduces incoming rainfall based on daily foliage area, accounting for allometry, competition, frost, phenology, and drought-defoliation effects. Canopy storage volume, derived from daily foliage area, accumulates rainfall and releases water through evaporation, with throughfall calculated using a simplified Beer-Lambert formula. Refer to Appendix L for more details and model equation.

## 2.4.4 Rooting system representation in PHOREAU

The explicit representation of root and their related processes is crucial for any model aiming to simulate the response of vegetation to climate change (Woodward and Osborne, 2000). Because of this, the framework for representing roots in PHOREAU had to be considerably expanded compared to the parent model where the rooting system was reduced to a simple fine root surface. In particular, we built upon the original SurEau framework (Cochard et al., 2021; Ruffault et al., 2022) by integrating coarse root depth alongside fine root surface, having the roots of different trees share the same soil to compete for water, and implementing plastic responses of root biomass and root depth to drought stress and aboveground growth.

The modelling of the root compartment in PHOREAU is based on the same major hypothesis as that of the canopy and light competition module: an implicit homogenous horizontal distribution of trees, with an explicit vertical stratification. In the same way the aggregated vertical distribution of foliage area entirely determines the light availability of each tree, competition for soil water between trees in PHOREAU is the result of the vertical distribution of their root systems. The underlying hypothesis is that all trees in a given patch compete for the same water reserves, provided their roots go deep enough; and the user must take care to select a patch stand area small enough to verify this constraint, which will itself vary according to the size and rooting structure of the trees present in the stand.






In PHOREAU the rooting system of a tree is split between fine roots and coarse roots: this distinction is essential as the root types have different functional roles and responses to external factors (Pregitzer, 2002). Schematically, fine roots extend horizontally to absorb water in the available soil, while coarse roots explore deeper layers and make them available to fine root exploration. Because in PHOREAU the soil is segmented in a number of layers, this has been translated in the following way: the fine root area of a tree in a given soil layer determines the conductance between this tree and the soil layer, while the rooting depth determines which layers the tree has access to. For a given rooting depth, fine root area is distributed between the soil layers following the negative exponential model presented in (Jackson et al., 1996), using a species-specific root distribution parameter.


In practice this means that, for a given set of soil parameters, certain trees are able to extract water from the full soil profile, while others are restricted to only a fraction (see Fig. W2, extracted from the PHOREAU evaluation on the ICOS sites). This framework is intended to reflect the crucial role of rooting depth in resilience to drought stress (Canadell et al., 1996), as trees with deeper rooting systems are able to make use of relatively untouched water reserves in deeper soil layers. Furthermore, because this is implemented in a forest dynamics model where many trees share the same soil, PHOREAU is able to use the differential rooting depths to explore the contrasting intra and inter-specific drought responses observed in nature (Johnson et al., 2018).



Rooting depth is a notoriously difficult trait to measure, and involves costly, time-consuming, usually destructive techniques (Maeght et al., 2013). While some rooting depth data is available in the literature (Guerrero-Ramírez et al., 2021), its scarcity makes it difficult to disentangle environmental, allometric, and genetic factors; what is driven by aboveground biomass, from what is driven by water availability and groundwater table depth (Fan et al., 2017; Freschet et al., 2021; Li et al., 2022). To circumvent this difficulty in obtaining accurate rooting depth traits,

we take advantage of the fact PHOREAU does not explicitly represent the position of a tree in the plot and ignores lateral distribution, by using coarse root biomass — an extensively studied trait — as a proxy for rooting depth, thereby implicitly aggregating the lateral and vertical extension of the root system in an integrative rooting extent variable, which is driven by shoot size and site aridity (Tumber-Dávila et al., 2022).

Coarse root biomass and fine root biomass in PHOREAU are calculated independently. Fine root area is derived on a 1:1 basis from leaf area. Meanwhile, coarse root biomass is calculated as the product to above-ground biomass with a root-shoot ratio, this root-shoot itself calculated as ratio of realized tree height to maximum species height, positively modulated by the mean of past drought indices (Morin et al., 2021). This formulation, shown in Eq. 30 to 32, follows the conclusions of Ledo et al., 2018 which identifies size and past droughts as the main factors driving root-shoot. These simple equations allow PHOREAU to capture several well-established characteristics of the evolution of coarse and fine root biomass.



$$R/S_{realised,s}^{n} = R/S_{min,s} + \frac{1}{2} * AllometryComponent + \frac{1}{2} * AdaptationComponent$$
 Eq.30

AllometryComponent = 
$$\left(R/S_{max,s} - R/S_{min,s}\right) * \left(1 - \left(\frac{Height^n}{Hmax_s}\right)\right)$$
 Eq. 31

$$AdaptationComponent = (R/S_{max,s} - R/S_{min,s}) * \sum_{i=(n-10)}^{n} (DroughtIndex^{i})$$
 Eq. 32

n: simulation year; s: species





Similarly to leaf shedding, fine root area tends to decrease in response to past drought events (Brunner et al., 2015; Hartmann, 2011). Meanwhile, total root biomass relative to aboveground biomass (the *root-shoot ratio*) has repeatedly been shown to be positively correlated to past drought events (Mokany, Raison and Prokushkin, 2006), and tree species adapted to more xeric climates have higher root-shoot ratios and deeper roots than those adapted to wetter conditions. These patterns, captured by PHOREAU (Fig. W17), are in accordance with Optimal Resource Partitioning theory (OPT), which predicts trees should increase their absorptive capacity relative to their transpiring surface under short water supply (Coomes and Grubb, 2000; Hertel et al., 2013).

Another observation captured by deriving root biomass from relative height in PHOREAU is the negative correlation between root-shoot ratio and above-ground biomass (Ledo et al., 2018; Mokany et al., 2006). Because tree height in PHOREAU tends asymptotically towards the species' maximum height following a parabolic curve, as trees become older they allocate proportionally more growth to their diameter than to their height — and to their roots in the new formulation. Following Konôpka et al., 2010, the maximum root-shoot was set to be greater for angiosperms than coniferous trees, who tend to have shallower roots (Schenk and Jackson, 2002) and less variation between juvenile and adult individuals. Another implication of this formulation is that the proportion of fine roots exponentially decreases with total root biomass (Li et al., 2003).

An emergent property of this framework is that for a given magnitude of water stress, a site which has already suffered past drought events will suffer less mortality and growth loss than a previously wet site, because of the

rooting depth adaptation mechanism (Fuchs et al., 2020). This type of plastic adjustment is concurrent with spatial variability in tree dieback related to the level of past drought acclimatization (Piedallu et al., 2023). Fig. W17 shows an example of this emergent behavior, by comparing simulations with two different climatic trajectories.

This integration of root plasticity, coupled with leaf shedding, is an important first step in the modelling of tree adaptation to drought conditions. However, it by no means provides a complete picture of the various strategies used by trees *in natura*. To refine our approach, the relative importance of past drought conditions relative to that of tree allometry in determining total rooting depth could be determined on a species by species basis, instead of a simple angiosperm/coniferous split. Even then, root plasticity is only one among many plastic responses to drought conditions: regulatory responses have been identified in the ectomycorrhizal network, non-structural carbohydrate concentration, differential gene transcription and pathways, increased suberin and lignin formation in roots, and decreased fine-root turnover rate (Bréda et al., 2006; Brunner et al., 2015).

#### 2.5 Model calibration and simulation initialization






*Species parameters*. Species parameters were not tuned on the basis of the evaluation datasets, and, for the majority, correspond to traits determined *a priori* from the literature and experimental results. A full list of the species parameters used in PHOREAU can be found in Table S13, with accompanying descriptions, examples, and data source references.

Site parameters. Site climatic and edaphic conditions were constructed using a mix of on-site measurements, and publicly available European datasets (see Sect. 3.1).

## 3 Model Evaluation



**Figure 4** | Framework for PHOREAU validation. In red the evaluation dataset (described in Sect. 3.1), in green the evaluated model outputs. Line direction indicate relative strengths and weaknesses of each method.

The key novelty of the PHOREAU model is that it is designed to predict a wide range of forest characteristics and ecosystem functioning features, occurring at various scales. Therefore, we evaluated the model across a broad spectrum of outputs, ranging from daily plant physiological measurements to long-term species composition predictions. This comprehensive approach allowed us to avoid one of the common drawbacks of earlier generation forest gap-models for temperate forests, which were often evaluated against high-level integrative metrics — such as long-term predicted total stand basal area, species distribution or potential natural vegetation composition (Botkin et al., 1972; Bugmann, 1996; Kienast, 1987) — which limits the robustness of their predictions under future conditions. By directly assessing the model's ability to reproduce intermediary variables, such as leaf area indices or soil water fluxes, we could control for common biases that may arise from errors offsetting each other under current conditions, which may not hold true when projecting into future climatic scenarios.

Depending on the targeted variable (and especially the available data to characterize it), the model evaluation was conducted on certain sites in France, or on many sites over Europe. Because PHOREAU is intended to be continuously improved and refined over time, the validation protocol and all associated data — summarized in Fig. 4 — will serve as a baseline to evaluate any future modifications to the model.

#### 3.1 Data sources

#### 3.1.1 ICOS sites

We used data from the Integrated Carbon Observation System (ICOS) for our most in-depth validation protocol that includes hydrological, growth, and mortality components. In particular, we selected four forested sites from the terrestrial ICOS Ecosystem network: Puéchabon, Font Blanche, Hesse, and Barbeau. Together these sites represent a diversity of the climatic, edaphic, and biotic conditions that can be found in France (Fig. 6). Refer to Appendix N for general details on the ICOS network.



870

875

A preliminary task was building an exhaustive database of all relevant input and output variables over the selected sites. This was made possible by the collaboration of each of the site PIs, especially for non-flux data that was not always readily available on the ETC database (Papale et al., 2006; Reichstein et al., 2005). Table 1 provides a summary of the ICOS data sources used in the model evaluation, as well as some of the main site characteristics, while a more in-depth description of each site can be found in supporting information (Appendices O, P, Q and R). Eventual gaps in data were corrected by selecting, for each of our four sites, the simulation period where the most harmonized data was available. Fig. 5 shows a simulated representation of the initial state of each inventory, highlighting the structural diversity across sites, and Fig. W1 a vertical representation of leaf area distribution.

|                                                            | Barbeau                                                                                                                                                                | Font-Blanche                                                                                                     | Hesse                                                                                                                                         | Puéchabon                                                                                                                        |
|------------------------------------------------------------|------------------------------------------------------------------------------------------------------------------------------------------------------------------------|------------------------------------------------------------------------------------------------------------------|-----------------------------------------------------------------------------------------------------------------------------------------------|----------------------------------------------------------------------------------------------------------------------------------|
| Location                                                   | 48°28′N, 2°46′E <sup>1</sup>                                                                                                                                           | 43°44′29″N, 3°35′45″E <sup>1</sup>                                                                               | 48°40′30″N, 7°3′59″E <sup>1</sup>                                                                                                             | 43°14′27″N, 5°40′45″E <sup>1</sup>                                                                                               |
| Altitude                                                   | 100 m above sea level <sup>1</sup>                                                                                                                                     | 425 m above sea level 1                                                                                          | 300 m above sea level 1                                                                                                                       | 270 m above sea level <sup>1</sup>                                                                                               |
| Simulation Period                                          | 2006-2021                                                                                                                                                              | 2007 - 2020                                                                                                      | 1999 - 2010                                                                                                                                   | 2003 - 2020                                                                                                                      |
| Simulation Patch Area                                      | 9 x 1000 m <sup>2</sup>                                                                                                                                                | 24 x 267 m <sup>2</sup>                                                                                          | 4 x 300 m <sup>2</sup>                                                                                                                        | 3x 100 m <sup>2</sup> (MIND control plots) 5                                                                                     |
| Available stand inventory data                             | Basal area aggregated by size and species <sup>s</sup>                                                                                                                 | Individual DBH measurements s                                                                                    | Individual DBH measurements <sup>s</sup>                                                                                                      | Individual DBH measurements <sup>5</sup>                                                                                         |
| Mean annual temperature                                    | 11.2°C 1,3                                                                                                                                                             | 14.8°C s                                                                                                         | 10.1°C s                                                                                                                                      | 13.6°C 1                                                                                                                         |
| Mean annual precipitation                                  | 677 mm <sup>1,3</sup>                                                                                                                                                  | 703 mm <sup>s</sup>                                                                                              | 948 mm <sup>s</sup>                                                                                                                           | 987 mm <sup>1</sup>                                                                                                              |
| Soil Description                                           | gleyic luvisol<br>millstone bedrock <sup>1</sup>                                                                                                                       | Silty clay loam<br>50%-90% rock fraction<br>Limestone bedrock <sup>2</sup>                                       | Luvic cambisol with local stagnic<br>tendencies<br>Deep loam clay layer <sup>1,2</sup>                                                        | Silty clay loam<br>75%-90% rock fraction<br>Limestone bedrock <sup>3</sup>                                                       |
| Maximum Available Soil Water<br>Quantity (over 5m profile) | 405.3 mm <sup>4</sup>                                                                                                                                                  | 178.4 mm <sup>s</sup>                                                                                            | 447.9 mm <sup>3</sup> (ESDAC prediction)                                                                                                      | 130 mm <sup>s</sup>                                                                                                              |
| Dominant tree species                                      | Sessile Oak ( <i>Quercus petraea</i> )<br>European hornbeam (Carpinus<br>Betulus) <sup>1</sup>                                                                         | Aleppo pine ( <i>Pinus halepensis</i> Mill. ) Holm oak (Quercus ilex L.) <sup>1</sup>                            | European beech (Fagus Sylvatica<br>L.)<br>European hornbeam (Carpinus<br>Betulus)<br>Silver birch (Betula Pendula)                            | Holm oak (Quercus ilex L.) <sup>1</sup>                                                                                          |
| Initial Basal Area                                         | 25.4 m <sup>2</sup> / ha <sup>s</sup>                                                                                                                                  | 19.6 m <sup>2</sup> / ha <sup>s</sup>                                                                            | 19.4 m² / ha s                                                                                                                                | 30 m <sup>2</sup> / ha <sup>3</sup>                                                                                              |
| Dominant Tree Height                                       | 25 m <sup>1</sup>                                                                                                                                                      | Pine: 13.5 m <sup>1</sup><br>Holm Oak: 5.5 m <sup>1</sup>                                                        | 18.3 m <sup>1</sup>                                                                                                                           | 5.5 m <sup>1</sup>                                                                                                               |
| Initial Stem Density                                       | 212 / ha <sup>2</sup>                                                                                                                                                  | 1008 / ha <sup>s</sup>                                                                                           | 3297/ha <sup>1</sup>                                                                                                                          | 4900 / ha <sup>3</sup>                                                                                                           |
| Stand thinnings                                            | 2011 : 15% of basal area <sup>s</sup>                                                                                                                                  | No                                                                                                               | 2005 : 25% of basal area<br>2010 : 15% of basal area <sup>s</sup>                                                                             | No                                                                                                                               |
| Leaf area index (LAI)                                      | 3.5 — 6.4 s, 2                                                                                                                                                         | 2.9 <sup>2</sup>                                                                                                 | 4.6 — 7.6 <sup>1</sup>                                                                                                                        | 2.2 2                                                                                                                            |
| Flux data                                                  | Provided by Site Manager 5,6                                                                                                                                           | Provided by Site Manager                                                                                         | Provided by Site Manager                                                                                                                      | Provided by Site Manager                                                                                                         |
| Tree water potentials                                      | Betsch et al. , (2011)<br>Peiffer et al. , (2014)                                                                                                                      | Provided by Site Manager                                                                                         | Provided by Site Manager                                                                                                                      | Provided by Site Manager                                                                                                         |
| References                                                 | 1: Delpierre et al., 2016 2: Briere et al., 2021 3: Davi et al., 2005 4: Maysonnave et al., 2022 5: Cuntz et Joetzjer, 2023 6: Reichstein et al., 2005 3: Site Manager | <sup>1</sup> : Monero et al., 2021<br><sup>2</sup> : Simioni, Marie and Huc, 2016<br><sup>5</sup> : Site Manager | <sup>1</sup> : Granier et al., 2008<br><sup>2</sup> : Dufrene et al., 2005<br><sup>3</sup> : Tóth et al., 2017<br><sup>5</sup> : Site Manager | 1: Limousin et al., 2011 2: Limousin et al., 2022 3: Rambal et al., 2014 5: Gavinet, Ourcival and Limousin, 2019 5: Site Manager |

**Table 1** | Selected stand characteristics for the four ICOS sites used in the in-depth PHOREAU validation, with associated data sources.

# 3.1.2 RENECOFOR and ICP II sites

To evaluate our model's predictions of tree and stand productivity, potential natural vegetations, and observed foliage areas, we used 250 plots spread across Europe, from 37.03° N to 69.58° N, and 8.17° W to 30.71° E, covering most of the major European species (Fig. 6. They cover a large range of environmental conditions, with mean annual temperatures (MAT) ranging from –1.62 to 17.6 °C, mean annual precipitation sum (MAP) ranging between 405 and 2707 mm, growing degree days (GDD) ranging from 475 to 4287 °C, and available water quantities ranging from 30 to 671 mm over the soil profile. Refer to Fig. 15 for the distribution of site abiotic conditions, and Table S2 for a detailed site by site breakdown.

The RENECOFOR network. Following the framework of the ForCEEPS validation (Morin et al., 2021), the RENECOFOR permanent forest plot network was used as the primary validation dataset (Ulrich, 1997). RENECOFOR makes up the French portion of the European ICP II network. Comprised of 102 plots (ca. 0.5 ha) in even-aged managed forests, each composed mostly of a single dominant species, they cover most of the main

tree species and environmental conditions in France — with the notable exception of Mediterranean conditions. From the year 2000 onwards, the plots were exhaustively inventoried every five years, as well as before and after every eventual thinning. After the removal of the plots that had suffered the strongest perturbations — and in particular the 1999 windstorm — 97 plots remained. With these, we constructed 192 testing datasets, by grouping for each plot between 2000 and 2021 every pair of inventories that were separated by a period of at least four years within which no disturbance was recorded. The mean initial basal area of the plots was 28.3 m²/ha, while the time-interval between inventories ranged from 4 to 15 years, averaging at 7.1 year. As a rule, we avoided longer time-lapses, which would have made disregarding regeneration and mortality more problematic, and would also have masked the model's performance in capturing the effects of yearly dynamics in productivity (which, for a model developed in the view of capturing the short and medium-term effects of climate change, is more important than representing mean aggregated past trends).

The ICP II network. In addition, we also used 148 plots from the International Co-operative Program on Assessment and Monitoring of Air Pollution Effects on Forests (ICP Forests), which comprises a network of intensively monitored forest sites (level II plots) distributed across Europe (de Vries et al., 2003; Schwärzel et al., 2022). These plots, located in various European countries, allowed the testing of the model over a wider range of abiotic and biotic conditions. This extension of the validation protocol was facilitated by the fact the RENECOFOR network is the French declination of the European-level ICP II program, with comparable protocols and measurements. Unlike for RENECOFOR, each plot corresponds to exactly one simulation dataset, with no repeat inventories separated by intervals of years. The mean initial basal area of the plots was 28.1 m²/ha, while the time-interval between inventories ranged from 2 to 10 years, averaging at 4.6 years (refer to Table S2 for details on each individual simulation dataset).

Figure 6 | Spatial distribution of sites used for PHOREAU validation. Sites are color-coded based on the dominant species identified in the inventory (see legend in top-left). Red-bordered diamonds represent the four ICOS site (Puéchabon, Font-Blanche, Barbeau, and Hesse) selected for in-depth hydraulic validation.

## 3.1.3 Climate and soil data





PHOREAU requires detailed daily climatic inputs, as well as comprehensive soil moisture retention measurements (see Table 1). To evaluate PHOREAU we used the ERA-5 Land dataset, a climate reanalysis providing various fields over the world at ~9km resolution (Muñoz-Sabater et al., 2021). The hourly data was aggregated to produce daily time-series from 1969 to 2021 over Europe for our study. Potential evapotranspirations were then calculated at the same resolution using the Penman-Monteith equation (Monteith, 1965).

PHOREAU requires, for each layer of soil (in this study 30 layers, up to a total depth of 5m, see Sect. 2.4.4), the fraction of coarse elements, as well as the parameters of the Van Genuchten water retention curve which describes the soil texture (Van Genuchten, 1980). These parameters were obtained for several depths from the European Soil Hydraulic Database (ESDAC) (Tóth et al., 2017), and interpolated over the height of the soil profile.

The resulting ESDAC soil and ERA-5-Land climate parameter files were used as a baseline for our European validation, and were directly used for the ICP II plots, for which no other climatic or soil data was available. When possible, we completed this continental-scale data with higher-resolution measurements. Field measurements were

available for all four ICOS sites, as well as for the RENECOFOR plots for which we used a combination of soil measurements and the SILVAE climate time-series (in particular fine-grain temperature and precipitation data that better account for site topography and exposition) to refine our validation. The mean-correction method used to integrate daily ERA-5 and monthly SILVAE climate time-series is presented in Appendix T. The workflow for climate reconstruction is summarized in Fig. 7.

On-site climate measurements were available for 26 of the 102 RENECOFOR sites (see Table S4 for the list of sites). For some of the sites the measurement periods only partially matched the simulation periods, while for others they were continuous from 2000 to 2021. These datasets, although not directly used in our evaluation protocol (so as not to bias our results for certain sites and species) were instead used to validate our climate reconstruction: first through direct comparisons of climate variable means and variances, and then by comparing the outputs of the ForCEEPS simulations carried-out with on-site *vs.* reconstructed climatic data (refer to Table S13).


960

955

Local measurements of SWHC were available up to a depth of 1 meter for all RENECOFOR plots (Brethes and Frankreich, 1997). Additional measurements were available up to 2 meters for more than half of the plots (Brethes and Frankreich, 1997; Lebourgeois, 2006; Guillemot, unpublished data), which were used to refine validation soil parameters.

970

Figure 7 | Summary of the workflow used for constructing PHOREAU evaluation inventories and climate datasets.

#### 3.2 Evaluation Protocol

## 3.2.1 Evaluation of intra-annual stand fluxes, tree hydraulics, and feedbacks on stand structure

For each of the four ICOS sites, model predictions were compared to observations at two distinct levels. First for stand-level structure, focusing on the annual trends of leaf area, basal area, and tree mortality, for which statistical metrics were not applied, but predictions instead served as a baseline to identify discrepancies between observed and predicted fluxes (but refer to Sect. 3.2.2 and 3.2.3 for direct evaluations on stand leaf area and productivity).

Second, for stand fluxes and tree functional dynamics, measured at the daily level. The performance of the PHOREAU model in reproducing the hydraulic functioning of forest stands was assessed for the following variables (from the most aggregative to the most specific): stand real evapotranspiration (ETR); evolution of soil water content (SWC); tree transpiration derived from sapflow; and stem water potential. Model performance was assessed using the Pearson correlation coefficient (*r*), the root mean square error (RMSE) and the mean deviation (MD) between observations and model predictions.

## 3.2.2 Evaluation against leaf area index

The evaluation of PHOREAU's ability to predict leaf area indices from inventories was realized on two different levels: first, by comparing model results to those obtained from satellite data for 340 sites spread over Europe featuring a large range of tree species; second, by comparing model results to LAI observations inferred from litter retrieval experiments for 40 sites in France.

The novelty of this kind of validation, as well as its importance when considering the fact PHOREAU predicts plants water use without any *a priori* fixing of foliage area (unlike most other tree hydraulics models), are presented and discussed in further detail in Appendix U.

The LAI satellite data used was retrieved from the Copernicus Global Land Service time series derived from daily PROBA-V satellite observations between 1999 and 2020 — first at a 1km resolution, then at 300m from 2014 (Fuster et al., 2020). For all RENECOFOR and ICP II sites and dates used for productivity validation (see Table S2) we compared LAI values predicted from the inventories at the start of the simulation, to those observed by PROBA-V and averaged over the summer months of the given year (but note these values are themselves uncertain (Fang et al., 2019) and likely underestimated for the denser sites).

LAI evaluation on litter data was restricted to those RENECOFOR sites where such data was available — mostly beech and oak sites, excluding coniferous-dominated stands not suited to litter retrieval (Ulrich, 1997).

## 3.2.3 Evaluation against productivity

For each of the 340 selected RENECOFOR and ICP II simulation plots, five patches of 1000 m<sup>2</sup> were initialized using the inventory of the first inventory campaign (see Table S2). For each patch, trees were sampled at random

within the first inventory, until the basal area per hectare of the simulated patch matched that of the original inventory. Sampling was done without repetition within each patch, but with repetition among patches. Trees that were absent from the second inventory or found dead were kept in the sampling in order to match simulated plots to real inventories, but were removed after for growth comparison. As the time step for validation was deliberately kept short, model mortality — either due to stress, age or density — were deactivated for this productivity validation protocol, so as to have for each sampled tree the observed and simulated final diameter. To evaluate the specific impact of the addition of tree phenology and hydraulics to the model, PHOREAU simulation results were compared against ForCEEPS predictions.

For tree species currently not parametrized for ForCEEPS (see Table S13 for a list of the 35 parametrized species), such as *Pyrus communis* or *Ilex aquifolium*, we used one of the generic sets of parameters. In addition to mortality, seedling regeneration was also deactivated in the model, due to the short time scales considered. The crown ratio between tree height and foliage height was initially set at the species maximum value, and initialized with the canopy bootstrap algorithm (see Fig. M1).

1020

1025

Simulations were run for each site over the time periods indicated in Table S2, repeated five times for each of the five sampled patches. We compared simulated and observed basal area growth at both the tree scale and the stand scale, using predicted and observed basal area increments (BAI) normalized to mean annual values. While comparing actual, instead of averaged, annual increments would have constituted a stronger test, this data is not available for size of plots and the range of species and climatic conditions considered here. For stand-level comparisons, results were directly averaged over the five patches. The performance of both the PHOREAU and ForCEEPS model were assessed using the Pearson correlation coefficient (r), the root mean square error (RMSE), the average bias (AB), and the average absolute bias (AAB) between observations and model predictions.

#### 3.2.4 Evaluation against potential natural vegetation

To evaluate the model's ability to predict forest composition through long term simulations for a broad range of climatic conditions — thus integrating the effects of all the different processes for mortality, reproduction, phenology, microclimate buffering effect, and competition not directly captured by shorter-term validations protocols —, we compared community compositions simulated by PHOREAU with the predicted potential natural vegetation (PNV) along an environmental gradient. Here, similarly as in Bugmann (1996) and Morin et al. (2021), potential natural vegetation is simply defined as the assumed dominant tree species, assuming no large disturbances, in a space spanned by mean annual precipitations (MAP) and mean annual temperatures (MAT), following Ellenberg (1986), Rameau et al. (2008), and San-Miguel-Ayanz et al. (2016). For this validation, we used the same 250 sites (RENECOFOR and ICP II) used for the productivity validation, spanning across all the different PNV conditions described in Ellenberg (1986) (Fig. 15).

For each of the 250 sites, we ran 2000-year simulations starting from the bare ground. This simulation length – accounting for seedling establishment, tree growth and mortality – was necessary to ensure the communities were no longer in a transient phase, and had reached the final stage of forest succession with a pseudo-equilibrium composition. The 2000-year climate time series was obtained by randomizing the years for which climatic data

was available (1969-2020), which preserved inter-annual variability in climate, but avoided any cyclic trend. For each site we considered 50 independent patches of 1000 m². At the end of the simulation, aggregate species basal areas per hectare were extracted for each simulated site and ranked from top to bottom. A simulation was classified as 'accurate' when the top ranked species belonged to the list of dominant species of the corresponding niche; as 'partially accurate' when the second ranked species matched, but not the top ranked; and as 'inaccurate' when neither the first nor the second predicted species belonged to the expected list.

## 4 Results Analysis

# 4.1 Evaluation of stand fluxes, tree hydraulics, and feedbacks on stand structure

The results of the in-depth evaluation of PHOREAU at the four highly instrumented ICOS sites demonstrated a good ability of the model of the model to reproduce observed ecophysiological and dendrological data across a wide range of scales. The model closely followed observed trends in stand basal area (average R² of 0.59, see Table S15), despite the inherent challenge of predicting individual tree mortality (Fig. 8). It accurately captured both the magnitude and variability of dieback across sites, in terms of both tree density (Fig. 9) and basal area loss (Fig. W4), with a marked increase in the rate of basal area loss in the latter years of each simulation; however, the model slightly overestimated mortality numbers on average and particularly at *Hesse* (+ 56%, see Table S17), as well as the share of large tree death relative to medium trees and saplings. Predicted foliage area results aligned well with observations in the two open evergreen sites with low mean deviations (*Puéchabon*: 0.19; *Font-Blanche*: 0.35, Fig. W3, Table S16). PHOREAU captured the quick regrowth in foliage area observed at *Hesse* after the 2005 cut (Granier et al., 2008); however, when comparing absolute values, PHOREAU noticeably underestimated foliage area in the two denser deciduous forests, consistent with prior validation results on leaf area (see Sect. 4.2). Despite these biases, the overall alignment between predicted and observed forest dynamics provides a solid foundation for comparing stand functioning and tree physiological responses at fine temporal resolutions.

The PHOREAU model predicted daily evapotranspiration (ETR) across the ICOS sites (and upscaled transpiration for Hesse), with relatively low mean deviations (*Puéchabon*: 0.03; *Barbeau*: –0.24; *Hesse*: 0.8) and good Pearson correlations (*Puéchabon*: 0.64; *Barbeau*: 0.79; *Hesse*: 0.62) between observed and predicted values (Fig. 10 and W5). At *Font-Blanche*, correlation was only moderate (r = 0.48, p 

1105

1110

Figure 8 | Predicted versus observed evolution of annual stand basal area. For each simulation site, the bars depict the annual basal area projections generated by the PHOREAU model, broken down by species and size class contributions (refer to Table S15 for associated statistics). The dashed line represents the observed annual total basal area derived from inventory data. Basal area is defined as the cross-sectional area at breast height of all trees per hectare.

Figure 9 | Predicted versus observed annual tree mortality. For each simulation site, the bars depict the total annual number of dead trees, irrespective of cause, broken down by species and size class contributions. Observed values are derived from stand inventories, while predicted values are generated by the PHOREAU model. Also shown are the annual mortality rates, calculated relative to the initial number of trees for two distinct time periods in each simulation, along with the total number N of dead trees by hectare. Transparent bars indicate years with thinnings (see Table S17 for details), which are excluded from the mortality statistics.

Figure 10 | Predicted versus observed evolution of monthly real Evapotranspiration (ETR) and tree Transpiration. For each simulation site (except Hesse, see below), the bars depict the monthly ETR (mm) predictions generated by the PHOREAU model, broken down by source of flux. Soil and intercepted water evaporation respectively originate from the first layer of soil and the water stored on the surface of leaves, while the two other sources are transpiration from different compartments of the PHOREAU tree (refer to Table S11 for details). The black points indicate the observed monthly actual evapotranspiration (with interpolated lines) representing the total water vapor released from the soil and vegetation into the atmosphere, aggregated from hourly or sub-hourly measurements obtained from each site's flux tower. For the Hesse site, tree transpiration has been upscaled from measured sap flux densities (site PI personal communication).

**Figure 11** | **Predicted versus observed evolution of soil water content (SWC).** For each simulation site, the black points indicate the observed daily actual SWC, with interpolated lines. The stacked bars depict the daily SWC (mm) predictions generated by the PHOREAU model, with individual contributions of each soil layer stacked and color-coded by soil layer (see Fig. W2 for layer details, and Table S12 for statistics). The predictions are confined the maximum measured depth for each site, as indicated in the upper right corner of the figure. For *Barbeau* and *Font Blanche*, observed SWC were directly obtained from site PIs; for *Puéchabon* and *Hesse*, they were interpolated from soil relative humidity (RH%) measured at different depths, using the same rock fractions as used in the simulation.

Figure 12 Aggregated predicted versus observed daily stem water potential. All available stem water potentials (mPa) observations plotted against PHOREAU predictions for the corresponding day and species. For each species, the full colored lines are the regression lines of the linear model of the relationship between observed predicted minimum water potential, with confidence interval represented with the grey dashed lines. The dashed red line is the 1:1 line. (a) Comparison with minimum water potentials. (b) Comparison with predawn water potentials.

# 4.2 Evaluation against leaf area index

Beyond local litter-based measurements, PHOREAU also demonstrated a reasonable capacity to estimate stand leaf area index (LAI) from observed data across many species and site conditions throughout Europe. When compared to PROBA-V satellite data (Fig. 13), the model yielded a good correlation between observed and predicted LAI values (r = 0.55, p 

Figure 13 | Projected (by PHOREAU) against observed satellite leaf area index (LAI) for all 340 RENECOFOR and ICP II validation inventories. The y-axis shows the LAI predicted by the model from the stand inventory at the start of the simulation, while the x-axis represents the PROBA-V LAI value for the maching coordinate and inventory year, averaged between July, August and September. Stand points are color coded by dominant species (see legend in bottom left). The size of points shows inventory basal area. The dashed red-line is the 1:1 line; the black full line represent the regression line of the linear model between observed and predicted LAI, with confidence interval represented by the grey shaded area. Associated statistics in Table S6.

### 4.3 Evaluation against productivity

At the stand level, PHOREAU exhibited robust performance in reproducing mean annual BAI across most species and environmental conditions. Overall, there was a strong correlation between observed and predicted values across all 340 simulations (r = 0.62, p 

**Figure 14** | **Projected (by PHOREAU) against observed mean annual stand basal increments (BAI)** for all 340 RENECOFOR and ICP II validation inventories. Stand points are color coded by dominant species (see legend above). The dashed red-line is the 1:1 line; other full lines represent the regression lines of the linear model between observed and predicted stand productivity, with confidence intervals represented by the grey shaded area (in black the overall regression; colored lines for species-specific regressions). Associated statistics for the global simulation in top left, while species-specific statistics can be found in Table S2.

#### 4.4 Evaluation against potential natural vegetation




When comparing the distribution of predicted dominant tree species after 2,000-year simulations along the environmental gradient covered by 250 sites across Europe (Fig. 15), the model performed well, with 62% of predictions accurately matching observed community compositions, and 24% partially accurate predictions (outperforming ForCEEPS' 43% accurate predictions). Yet, PHOREAU's ability to accurately predict potential natural vegetation (PNV) varied depending on site conditions, with a noticeably larger uncertainty for Mediterranean forest types, humid beech forests, and mixed montane spruce-beech forests. A detailed view of the predicted dominant species (Figure W15) revealed that much of this uncertainty stemmed from PHOREAU's tendency to overestimate the competitive advantage of *Q. robur* relative to Q. petraea and F. sylvatica in both hot and mild climates. Despite these discrepancies, the model demonstrated strong predictive performance in extreme environments, accurately predicting species composition at both extremely cold and extremely warm sites.

tested sites in the PNV diagram of supposed dominating species (built according to mean annual temperature and precipitation sum). PNV dominating species are Pc (P. cembra); Pa (P. abies); Aa (A. alba); Fs (F. sylvatica); Qp (Q. petraea); Qr (Q. robur); Pp (P. pinaster); Ph (P.halepensis); Qi (Q. ilex) Circle colors indicate the agreement between simulated and PNV dominating species after the 2000 years PHOREAU simulations. Green: sites for which the dominating species (by basal area) was accurately predicted. Orange: sites for which the second-ranked species was accurately predicted, but not the firstranked. Red: sites for which neither the first-ranked nor second-ranked species were accurately predicted.

**(b)** Geographical repartition of the 250 sites (RENECOFOR and ICP II) used for PNV validation, colored by potential niche composition (matching the background colors of Fig. 15a). Shapes indicate prediction success, as described above.

### 5 Discussion









# 5.1 A process-based model to investigate diversity-productivity and diversity-resilience relationships

The difficulties inherent in integrating trait-based processes in a semi-empiric framework justified evaluating PHOREAU on a variety of metrics — including predicted foliage area, soil water and stem water potentials which, to our knowledge, has never been attempted before in an individual-based gap model (but see Eller et al., (2020), Xu et al., (2021), for cohort-based approaches). Furthermore, the bottom-up approach we have adopted mitigated the risk of error compensation and of equifinality, which often appear when some parameters or processes covariate and compensate each other in respect to an integrative metric. Avoiding equifinality was crucial to the development of PHOREAU, because as climatic conditions deviate from the historical baseline in future years, correlations between processes that were equifinal for historical conditions may shift, limiting the ability of the model to accurately predict the impact of climate change on forest functioning. While direct validation on annual growth is rarely done for gap models because of the inherent difficulty of reproducing such metrics for models not originally designed to work at such short temporal scales (Fyllas et al., 2014; Mette et al., 2009), the more granular representation of stand functioning of PHOREAU justified our evaluation on short-term individual tree and stand productivity. The good performance of the model across the wide range of species and conditions used in the productivity and PNV validation — including Mediterranean and boreal forests demonstrates its widespread applicability to European forest ecosystems. Furthermore, the state-of-the-art validation dataset used in this study will serve as a baseline to assess any further refinements to the model, as additional species traits become available.

In contrast to ecophysiological process-based models than can be parametrized using only physiological and functional traits (Davi et al., 2005b; Maréchaux and Chave, 2017), PHOREAU eschews a direct representation of carbon assimilation and allocation, in favor of a growth-reduction based approach. While this simplification does distort actual tree functioning and ignores the importance of carbon reserves in buffering year-on-year growth (Körner, 2003), it presents a number of advantages when considering the ecological processes that shape species composition. In addition to a significant gain in computing time, it curtails the uncertainty in model predictions that can result from equifinality, by limiting the number of variables directly impacting growth. However, by incorporating detailed physiological representations of few selected mechanisms such as tree phenology and hydraulics, we maintain the model's ability to react to shifts in climatic conditions, thereby striking a balance compared to more simplified representations. Furthermore, by calculating tree growth, leaf area, mortality and establishment rates on the basis of well-established observed parameter values, to which process-based reductors are subsequently applied, we were able to maintain realistic stand basal and foliage areas over the length of the simulation. This result is a prerequisite to any temporal exploration of diversity-resilience relationships in droughtstressed forests: only by accurately predicting the evolution of forest foliage and basal area can we then study the effects of species-mixing (Forrester and Pretzsch, 2015) for forests functioning at eco-hydrological equilibrium. This is why our integrative validation on the ICOS sites is an important milestone in the development of hydrologybased forest models: unlike usual hydrological validations (Morales et al., 2005), not only did PHOREAU provide robust predictions of water fluxes for many years over a diverse set of conditions and species, it did so with no a

*priori* fixing of stand leaf and basal area, instead calculating the evolution stand structure on the basis of water-stress feedbacks.

#### 5.2 Limitations and future avenues of improvement









Despite good correlations and low average bias, PHOREAU predictions consistently underestimated the observed variability across almost all considered metrics, including soil water quantities, stem water potentials, tree productivity, stand productivity, and stand foliage areas. This attenuating effect is in itself not surprising given the necessary simplifications presented by any modelling approach, and results from a number of unavoidable factors: precision of climatic and soil texture data (especially for ICP II sites); utilization of single sets of species parameters disregarding intra-specific genetic and phenotypic trait variability; lack of 3D representation of competition among trees. While climatic, soil, and species traits inputs can easily be refined for more granular simulations at the local and regional level, taking into account site exposition and fertility, the strong hypothesis of the PHOREAU model regarding the horizontal homogeneity of competition for light and water inside a patch will always be an obstacle to capturing the individual dynamics of trees advantaged or disadvantaged by microtopography and spatial allocation of tree crowns and rooting systems. Despite this inherent limitation, the integration in PHOREAU of many previously disregarded or implicit processes, including explicit roots, phenology, process-based tree hydraulics, and microclimate, has allowed it to outperform the ForCEEPS model in better predicting both short-term growth and long-term species composition. Furthermore, the gap between the two models' predictions is likely to become greater under future conditions, where PHOREAU is expected to be more robust as it explicitly represents key processes, such as drought stress and phenology, in a more mechanistic way.

However, by introducing a more granular representation of tree functioning, PHOREAU has induced a mismatch between some of the parameters used in the model and the role they were originally intended and calibrated for. This mismatch, particularly evident for the optimal species growth rate parameter ( $g_s$ ) and for foliage allometry parameters, is responsible for the difficulty in reproducing the growth of extremely productive trees, and the overall underestimation of the productivity of species like P. halepensis, F. excelsior, or A. pseudoplatanus (see Table S4). Because the optimal growth rate in ForCEEPS was calibrated for the main French species based on the top  $10^{th}$  percentile of annual diameter increments measured in the NFI database (IGN, 2020) and for other species dates back to even earlier studies (Didion et al., 2009), it is in reality more akin to a growth rate under relatively unconstrained conditions than an actual optimum. As we updated the model's representation of light and water use constraints to a more process-based approach, we have likely introduced constraints already implicitly present in this aggregated growth rate parameter, essentially penalizing trees twice for the same factor. As we continue to refine the PHOREAU model, a major challenge will therefore be recalibrating this parameter to better reflect actual potential growth unconstrained by competition, despite inherent difficulties in obtaining such data (Pretzsch, 2009).

Similarly, the parameters with which foliage area is derived from tree diameter have not been fully updated to reflect the new importance of foliage area in driving modelled water fluxes. Despite the many changes introduced in the representation of tree crowns and the partial validation on satellite data, the model demonstrated a poor

ability to predict measured litter LAI for sites of similar composition and basal area. Furthermore, neither satellite nor litter-derived total LAI measurements can be used to properly evaluate the predicted vertical distribution of leaf area. However, predicted vertical LAI distribution, from which microclimate and individual light-competition constraints are derived, is key to model ecological processes, and should therefore next be examined and validated against ground or airborne LIDAR and microclimate measurements.

Another obvious area of improvement for the model will be a deeper integration of the plant phenology component with other modelled processes. In this study leaf unfolding, leaf senescence, and probability of fruit maturation were computed yearly for an average individual of each species. This method captured inter-specific differences in phenology and temporal light partitioning, but did not account for intra-specific shifts in phenology caused by stand structure. By integrating model variables like microclimate, light availability, and water stress as inputs for an individual-based phenology calculation, PHOREAU will be able to capture well-established variations in leaf phenology between trees of different sociological status (Augspurger and Bartlett, 2003; Cole and Sheldon, 2017; Gressler et al., 2015; Schieber, 2012), which are responsible for the persistence of shrubs and saplings in mature forests (Gill et al., 1998; Vitasse, 2013). However, this development would require an independent validation of the recalculated phenological outputs; and forest inventory datasets with measurements of both plant phenology and understory microclimate are still few and far between.

## 5.3 Applications and future research perspectives









### 5.3.1 Establishing baseline available water: retro-engineering PHOREAU to predict rooting depths

One of the main causes for the model's attenuation of variability in stand and tree productivity was the uncertainty regarding the actual quantity of soil water available to the trees. This uncertainty is itself the result of a twofold gap in information: lack of data for the texture of deeper soil horizons, and the extremely simplified framework used to estimate tree rooting depths. By choosing to reduce the wide observed differences in rooting depths across biomes (Canadell et al., 1996; Fan et al., 2017; Schenk and Jackson, 2002) and species (Fan et al., 2017; Sperry et al., 2002) to a simple equation based only on tree size and an aggregate drought index based on past climatic conditions, we intentionally avoided any integration of model results (such as tree foliage area or percentage of embolism) in the calculation of rooting depths, as this would have resulted in an optimization of soil available water on precisely the variables we were trying to validate. Unlike other process-based models validated on stand hydraulic fluxes (Ruffault et al., 2023), the fact that PHOREAU produced robust multi-year predictions without using observations to control for stand leaf areas, rooting depths, or actual available water, confirms its possible applications to making realistic dynamic predictions across a large range of forests where this data is not available.

To overcome difficulties related to the soil water parametrization, an alternative approach could be used. For instance, based on the hydrological equilibrium hypothesis (HEH), which states that, in a given edaphic and climatic environment, trade-offs between vegetation water use and drought stress drive canopy density and forest composition toward an optimal hydric state (Caylor et al., 2009; Eagleson, 1982), and following the well-substantiated hypothesis that trees function near the point of catastrophic hydraulic failure with narrow safety margins (Choat et al., 2012; Tyree and Sperry, 1988), a retro-engineering of PHOREAU could be realized where

rooting depths are calculated by optimizing tree available water such that, for a given inventory and soil profile (Kirchen et al., 2017), foliage area is maximized (Grier and Running, 1977), and plant minimum water potentials are constrained to values to the point of catastrophic xylem failure. Compared to similar HEH-based statistical (Nemani and Running, 1989) or process-based (Cabon et al., 2018) modelling approaches, this retro-engineering of PHOREAU will natively integrate many inter- and intra-specific niche and competition processes that are integral to forests' actual water use. It will furthermore be a necessary first step in establishing a historical baseline when using the model to predict the medium-term impact of global change on forest composition and functioning, as available water is a major determinant in predicting drought-induced die-off events (Allen et al., 2010).

# 5.3.2 Unraveling the effects of trait diversity on competition and coexistence









The novel approach presented in this study, integrating plant functional traits in a forest dynamics model, was developed to improve the generality of the calibration for new species, but also to cope with the difficulties encountered by ecologists when testing hypothesized links between trait diversity, species competition and coexistence. While differences in traits governing resource use should, intuitively, translate into niche differences that maintain coexistence through competition reduction, attempts to directly link trait dispersion with historical species coexistence have proven challenging (Adler et al., 2013; McGill et al., 2006). This challenge arises from the fact most traits impact competition for several resources at the same time, and that even a temporary advantage in growth can actually result in a lower global fitness when considering population dynamics, with for example feedbacks on drought-induced mortality (Forrester and Pretzsch, 2015) or frost damage due to early onset leaf unfolding (Bigler and Bugmann, 2018). To overcome this difficulty, process-based models of resource competition with processes explicitly relying on species traits have been proposed as a way to unravel the mechanisms linking trait diversity to forest functioning (Levine et al., 2024). Because the effects of climate change on forests will likewise be mediated by complex species mixing effects, the need to develop mechanistic models that bridge the gap between trait-based and ecology and empirical modelling has become urgent to assess the short and mediumterms effects of global warming on existing forests, and discriminate between the possible management scenarios available to forest managers.

The PHOREAU model, having been directly evaluated for most of its processes, could be used as a relevant tool to identify thresholds conditions for species coexistence, dominance, or extinction. A first parsimonious approach could simply consist in identifying the main processes — phenology, water-use, or competition for light — limiting a species fitness at the edges of its predicted distribution (Morin et al., 2007). A more involved exploratory protocol could follow the methodology outlined in Levine et al. (2024). By considering predicted species compositions for a wide range of climatic and edaphic conditions, and taking care to distinguish, for each set of condition, the different mechanistic processes which make up a species' competitive fitness, we could establish relationships between aggregated model metrics (for example growth reductors) and underlying species traits. These relationships could then be used to predict the impacts of climate change on forest composition. In parallel to this approach, and as a prerequisite, predicted species compositions should be compared to actual observed compositions, albeit for a much greater set of points than those for the potential composition validation presented in this study, dissipating any remaining uncertainties regarding the representation of regeneration and mortality, which is one of the main current challenges for forest modelling (Cailleret et al., 2017; Vanoni et al., 2019).






## 5.3.3 Evaluating management policies under future climate scenarios

A further policy-relevant application of the PHOREAU model in the coming decades lies in its ability to simulate forest management scenarios under different climate trajectories, and evaluate their outcomes based on key ecosystem service metrics, including wood production, biodiversity conservation, and carbon sequestration. As forests play an increasingly critical role in helping countries meet sustainable development goals (Chapin III et al., 2008), and with forests storing roughly half of terrestrial carbon (Friedlingstein et al., 2019), predicting forest carbon dynamics and its response to management decisions under climate change has become an essential consideration for forest managers. However, while policy makers — supported by the recorded increase in the European forest carbon sink in the early 21st century (Pan et al., 2011) — table on a continued increase in the share of carbon emissions removed by forests (with a target of 40% in France by 2050), this dynamic has already shown signs of slowing (McDowell et al., 2020) as the early forcing effect of climate warming on forest productivity is now counterbalanced by increased drought-induced tree mortality (Allen et al., 2010; Hammond et al., 2022). While previous studies have evaluated the performance of different management strategies for carbon sequestration over the next decades based on a priori global forest biomass trends and management rules (Bastick et al., 2024; du Bus de Warnaffe and Angerand, 2020), very few models, to our knowledge, have attempted the dynamic integration of forest management with stand-specific future conditions to predict the evolution of the forest carbon stock. By integrating management, growth, and hydraulic processes, PHOREAU is uniquely positioned to simulate more realistic and agile forest trajectories, and to help forest managers by giving them insights about how to better adapt forest to new environmental conditions through management actions.

In conclusion, by combining a detailed representation of plant functional traits with the flexibility required for large-scale simulations and species calibration, PHOREAU offers a unique compromise between ecophysiological realism and operational applicability — making it a valuable tool for both ecological research and forest management under climate change.

### Supplementary information

## Appendix A: Decoupling tree height from diameter: light-dependent plasticity

The predictive power of gaps models is tied with their representation of stand structure. Yet most classic gap models, including ForCEEPS, do not simulate a dynamic tree height, instead inferring it from the tree trunk diameter through an allometric relationship. It follows that for a given species, every individual follows the same height-diameter trajectory. While this is consistent with the fact most forestry surveys report basal diameter without height, this means that the models cannot represent site effects on maximum height, as well as the effects of competition for light on the height-diameter relationship. In reality dominated understory trees tend to allocate more carbon to height growth than diameter growth. Conversely, trees in low-density or thinned forests have greater diameter growth and slower height growth (Oliver and Larson, 1996). Furthermore, this sensitivity of growth allocation to competition for light is more marked in shade-intolerant species (Delagrange et al., 2004).

The effects of competition for light on growth allocation are crucial for understanding stand dynamics, as small initial differences in height tend to increase with time unless corrected by greater height growth. Forest managers have long known that tree maximum height varies from site to site with tree age and density (Fortin et al., 2019), and forest growth models often use different height-diameter depending on site conditions (Mehtätalo et al., 2015). Attempts to implement dynamic height growth in gap models have been shown to increase the realism of simulated stand structure, without reducing general applicability. For instance Rasche et al. (2012) have implemented such a dynamic height in the ForClim model on which ForCEEPS is originally inspired. Instead of the static relationship between diameter and height (h), height increments are calculated at each time-step  $\Delta H = f_h \Delta D$  through a function  $f_h$  that distributes growth between diameter and height growth according to a competition-for-light driven parameter s, which replaces the original fixed species-specific allometric parameter. Since the yearly diameter increment uses previous-year height in its calculation, its formulation also had to be adapted to account for the fact that height is dynamic and no longer directly calculated from diameter. These adaptations have been used in our modified ForCEEPS model, albeit with two important modifications.

Firstly, the parameters of the growth-distribution coefficient  $g_s$  were adapted to be more conservative, and better reflect the species-specific relationship that had already been parametrized:

1430 
$$f_h = s \times \left(1 - \frac{H - 137}{H_{max,s} - 137}\right)$$
 Eq. A1

$$s = s_{original} + (kLa * 10) * (1 - AL_H)$$
 Eq. A2

where kLa is the species shade-tolerance, H the tree height in centimeters,  $H_{max,s}$  the maximum species height, and  $AL_H$  the light availability at the top of the tree crown.

Secondly, we adapted the yearly growth equation. In the original formulation by Rasche et al. (2012), because yearly growth is calculated on the basis of total diameter at the start of the year, a tree that allocated more growth to height than to diameter due to competition in year n would have less total growth for year n+1 than a tree that had allocated more growth to diameter, all else being equal. This is a result of the simplifications of the ForClim model, in which the diameter increment is calculated on the basis of previous year diameter instead of the previous year volume. This means tree biomass is only dependent on tree diameter, disregarding its height. This effect has major implications, as originally taller but thinner trees end up with smaller final height and diameters than in the original formulation. A possible solution would have been to replace trunk diameter by volume in the growth equations; but this would have meant reshaping the model from the ground up, and making it less applicable to classic forestry datasets, as actual volume data are very rarely available. In the end, we adopted an ad-hoc solution by giving each tree two sets of heights and diameters: a static set ( $D_{static}$  and  $H_{static}$ ), calculated from the old equations and static allometry relationships, that were only used as an ad-hoc proxy for real tree volume in the updated diameter increment equation (Eq. A3) and the calculation of slow-growth mortality (to avoid killing off trees that allocate too much growth to height); and a real set (D and H), using the updated equations and dynamic allometry, that was used in all other cases including the light-competition module.

$$\frac{\Delta D}{\Delta t} = kG * D_{static} * \frac{\left(1 - \left(\frac{H_{static}}{H max}\right)\right)}{2 * H_{static} + f_h * D_{static}}$$
 Eq. A3

## Appendix B: Crown-length reversion

The dynamic change of tree crown length was modified to better represent the feedbacks between stand structure and competition for light. In PHOREAU, light availability impacts growth directly and indirectly: directly through the shading growth reduction factor, and indirectly through the crown-length growth reduction factor, which represents long-term crown shrinking due to shading. Individual tree crown lengths are calculated as the product of tree height, and a variable ratio that depends on species characteristics and tree status. This ratio changes according to the light exposition of the tree, between two extreme species-specific values as described in Morin et al. (2021). In the original ForCEEPS framework, seedlings started with a crown ratio set at the species maximum, which then decreased over the tree's lifetime with shading. In particular, this formulation assumes that the crown ratio can only ever decrease or stay the same from one year to the next, with no possibility of reversion when more light becomes available.

Therefore, we have implemented the possibility of crown ratio reversion in PHOREAU. A constantly decreasing crown ratio assumes no increase in light availability over a trees lifetime, disregarding the impact that the death or removal of one tree can have on its neighbours by enhancing light availability and leading to larger crown sizes and denser canopies (see Juchheim, 2020, and Saarinen et al., 2022). We have consequently adapted the original ForCEEPS crown ratio equation to reflect this, with a yearly increase capped at 5% of the difference between the previous-year crown ratio, and the potential crown ratio given current light availability. We are aware this approximation does not take into account the fact that younger trees recover their crowns better due to having more remaining growth potential (Hynynen, 1995).









## Appendix C: Species-dependent crown shapes

An accurate representation of crown shapes is an integral component to any model of light competition and canopy interactions between trees (Krůček et al., 2019). In reality the crown shape of any given tree is a complex combination of genetic, allometric, and environmental factors, as crown shape varies across species, age groups, climate, local conditions and the shading status of the tree (Oliver and Larson, 1996). Canopy packing in mixed forests can be partly attributed to this heterogeneity and plasticity of crown shapes, as trees suffer relatively less competition for a given foliage density (Longuetaud et al., 2013).

Crown-shape representation in PHOREAU iterates on the ForCEEPS framework, which already allowed for stratified distributions of foliage area over a vertical axis (Morin et al., 2021). Compared to the previous iteration, PHOREAU allows trees to have other crown shapes than the default inverse-cone – such as conical or ellipsoidal shapes. This is meant to represent broad patrons in crown geometry observed at the European Scale, such as the fact species present in higher latitudes or latitudes tend to have more columnar or conical crowns to capture light coming from a perpendicular angle, whereas species as lower latitudes are more frequently flat-topped for maximum exposure (Kuuluvainen and Pukkala, 1989).

While the lack of explicit tree positions prevent PHOREAU from recreating the asymmetrical crown shapes which result from horizontal constraining between crowns (Niklaus et al., 2017), this simple approach allows for a more accurate representation of side-shading between trees, and captures the way shaded trees tend to become more flat-topped as they reduce their crown height (Oliver and Larson, 1996), while saving some simulation time. See Figure 5 for a visualization of the new crown shapes.

# Appendix D: Density-dependent light availability

Any representation of forest canopies and light dispersion has to strike a balance between predictive power — how much photosynthetically active radiation (PAR) does a given tree actually receive at a given moment in time? — and computing cost: by aggregating leaves on a tree-by-tree basis and disregarding differences in angle and

— and computing cost: by aggregating leaves on a tree-by-tree basis and disregarding differences in angle and light absorption between sun and shade-leaves (Givnish, 1988), by calculating at yearly time-step, and by considering only the vertical stratification without an explicit representation of trunk distribution across space, ForCEEPS is able to compute in a timely fashion what would otherwise take orders of magnitude longer with a

more bottom-up approach from the leaf to the tree.

PHOREAU does not diverge from this general framework, which is well suited to working on large-scale inventories (that usually come without tree-level coordinates), and does not suppose any *a priori* knowledge on canopy composition. However, this simplification is not without its drawbacks. Because the light availability of a given canopy layer depends solely on the foliage area present in the layers above it, with no accounting for how this foliage is actually distributed, light competition is — in effect — boiled down to a single value: the LAI. Intuitively we understand that this does not quite tally with reality: two superposed leaves will intercept less light, all else being equal, than two leaves on a level plane; forests are not horizontally homogeneous, and gaps in the

canopy may form as trees die off, allowing saplings to sprout and grow even in dense stands (Nicotra et al., 1999). Due to the links between patchy structures of light availability and tree species diversity and coexistence (Moora et al., 2007), measuring and quantifying microsite light availability has been a focus of research (Parent and Messier, 1996; Tymen et al., 2017), with important implications for forest management (Coates et al., 2003).







This structural limitation — which can be important, e.g. to accurately predict species richness in relation to management — can never be fully worked around. And, in keeping with the general philosophy of the model to strike a balance between complexity and genericity, we opted not to incorporate a complex 3D tree-level light absorption model (le Maire et al., 2013). However, in the transition from ForCEEPS to PHOREAU, some steps have been taken to at least partially account for the horizontal stand structure. This was done in an indirect way by using information available to the model: the stand density.

As in most gap-models, foliage area in ForCEEPS is translated into light availability using a modified logarithmic Beer-Lambert law, see Eq. D1, where light availability is a function of foliage area and a light extinction coefficient  $\lambda$ . In the original formulation of the law this extinction coefficient is calculated by integrating over the path of the light ray the absorbance and density of the materials it crosses. This calculation — which accounts for the angle of the leaves, the angle of the sun's rays, the different absorbances between species and sun and shade-leaves, and the distribution and clumping of the leaves and trees (Dufrêne and Bréda, 1995; Smith, 1993) — is usually simplified into an empirical constant extinction parameter, which can vary from site to site (Binkley et al., 2013; Vose et al., 1995). However, in the ForCEEPS framework, where stand composition is an emergent property and not an input, a single  $\lambda$  value is used regardless of site conditions.

Following the methodology outlined in (Nilson, 1971; Black et al., 1991; Bréda, Soudan and Bergonzini), PHOREAU integrates a clumping factor  $\Omega$  in its calculation of the light extinction coefficient. This clumping factor ranges from 0 (corresponding to a fully concentrated distribution) to 1 (corresponding to a perfectly homogenous distribution), and represents the aggregation of leaves within each tree and between the trees themselves. The advantage of this approach is that  $\Omega$  can be calculated each year as an emergent variable, allowing the model to capture observed trends like the inverse relation between LAI and the light extinction coefficient (each additional increment of leaf area blocks marginally less light) (Dufrêne and Bréda, 1995). The clumping factor in PHOREAU is calculated using Curtis relative density (Smith, 1993; Curtis, 1982): with this formulation (see Eq. D2) for a given LAI, a dense stand with small trees will block out more light than a stand populated by a few large trees. This approach is similar to the one used in LAI estimation with MODIS or hemispherical photography, where clumping indices are also used to correct the raw measured LAI (Chen et al., 2012; Demarez



et al., 2008; Zhu et al., 2018).

A further step would be to incorporate species-specific absorbance values, as leaves of different species react differently to incoming light (Binkley et al., 2013), but this would necessitate gathering data at the species level (data which is, to our knowledge, available only for a select few species). Another possible refinement would be to incorporate the angle of incoming light in the calculation of light availability (Smith, 1980); but this would require modifying the light competition calculation to consider site effects related to slope and exposition.

$$Light\ Availability_{i} = e^{-\lambda * \sum_{j=i+1}^{n} (FoliageArea_{j})}$$
 Eq. D1 
$$\lambda = 0.25 * \frac{BasalArea}{\sum_{j=1}^{N} (DBH_{j}^{2})}$$
 Eq. D2

n: total number of layers; i: layer rank; N: number of trees








#### Appendix E: Incorporation of Specific Leaf Area

The relation between trunk diameter, crown biomass, and foliage area in ForCEEPS are governed by a set of simple allometric relationships calibrated for a few of the main temperate European species, using experimental data collected in Switzerland by destructive sampling in the 1940s and 50s (Bugmann, 1996; Burger, 1951). The refinements that ForCEEPS implemented regarding crown plasticity and explicit vertical stratification were built upon this foundation but did not challenge its underlying assumptions (Morin et al., 2021). This became problematic as the model — and PHOREAU in particular — incorporated more species from a larger geographic range: understory or Mediterranean species in particular that were not represented in the initial calibration dataset. This was directly reflected in model predictions, for example with an overestimation of *Quercus ilex* or *Pinus halepensis* mortality due to inflated foliage areas.

A simple solution to this issue was implemented by recalculating the  $c_2$  parameter (used in ForCEEPS to derive a tree's foliage area from its diameter) using a specific leaf area (SLA) value for each species. The retained SLA — the surface area for a given mass of leaves — are those of average adult individuals of each species over a large set of sites (Devresse et al., 2024; Kattge et al., 2020). This new formulation (see Eq. 22) allows the model to capture inter-specific differences in drought resistance strategies (Greenwood et al., 2017), while disregarding for the moment SLA plasticity to tree age, competition, and site conditions (Gratani, 2014).

#### **Appendix F: Microclimate derived from stand-structure**

By integrating fine hydraulic and phenological mechanisms in the overall framework of a forest-structure gap model, PHOREAU has the opportunity to capture the effects of microclimate on plant functioning. Because forest canopies absorb or reflect the majority of incoming solar radiation, reduce wind speeds, convert solar energy into latent heat through evapotranspiration, and block outgoing infrared radiation, climatic conditions in the understory are often buffered compared to those at the top of the canopy, with cooler more stable temperatures during the day, and warmer temperatures during cold nights. This climate dampening effect is more marked for temperature extremes, and for tall, structurally complex dense canopies (De Frenne et al., 2021). Furthermore, it is an important factor in ability of young, understory trees to resist droughts despite their shallow root systems (Forrester and Bauhus, 2016). Because PHOREAU evaluates drought-stress at an individual level by calculating tree fluxes, it can easily make use of microclimatic data for temperature, air humidity, and light availability, to better compute plant evapotranspiration and in turn differentiate water stress among individuals of different heights. In addition, because PHOREAU simulates many small patches each sharing a soil and a canopy height profile, the incorporation of microclimate could help the model capture forest landscape mosaic dynamics, where forests with

heterogeneous patches are able to host more diversity due to differentiated microclimatic effects on regeneration and drought (Pincebourde et al., 2016).









To derive microclimate temperature and air humidity from macroclimate, we implemented a version of the statistical model developed, calibrated and validated in Gril et al., (2023) and Gril, Laslier, et al., (2023). This model, which has the advantage of using only easily available patch characteristics, uses a simple slope and equilibrium approach, presented in Figure S1, to compute microclimate temperature at soil level  $(T_0)$  from the corresponding hourly or daily macroclimate temperature  $(T^j)$ . The slope  $(m_{slope})$  captures the linear relationship between microclimate and macroclimate, while the equilibrium is the point at which microclimate is equal to macroclimate (Eq. F3). In our case, month mean temperature  $(T^m)$  is used as the equilibrium. The slope, which acts as a buffer if is lower than 1, is computed daily using patch-level leaf area index (LAI), maximum tree height  $(h_{max})$ , and vertical complexity index (VCI), as seen in Eq. F4 with corresponding coefficients calibrated over a large dataset of microclimate measurements (Gril, Laslier, et al., 2023). VCI is obtained following Van Ewijk, Treitz and Scott, (2011) by calculating the weighted logarithmic average of foliage area proportion per patch canopy layer  $(p_i)$ , normalized by the total number of layers n, as shown in Eq. F5 and Eq. F6. Finally, for any given tree height h, the corresponding microclimate temperature  $T_h^j$  is derived from soil microclimate and macroclimate using a linear interpolation, as shown in Eq. F1 and Eq. F2.

$$T_h^j = T_0^j + (1 - (w(h)) \times (T^j - T_0^j))$$
 Eq. F1

$$w(h) = \frac{(h_{max} - h)}{h_{max}}$$

$$T_0^j = T^j \times m_{slope} + T^m \times (1 - m_{slope})$$

$$m_{slope} = e^{(0.39 - 0.04 \cdot LAI^j - 0.2 \cdot VCI^j - 0.07 \cdot h_{max})}$$
Eq. F3
$$Eq. F4$$

$$T_0^j = T^j \times m_{slope} + T^m \times (1 - m_{slope})$$
 Eq. F3

$$m_{slope} = e^{(0.39 - 0.04 \cdot LAI^{j} - 0.2 \cdot VCI^{j} - 0.07 \cdot h_{max})}$$
 Eq. F4

$$VCI^{j} = -\frac{\sum_{i=1}^{n} p_{i} ln\left(p_{i}\right)}{ln\left(n\right)}$$
 Eq. F5

$$p_i = \frac{FoliageArea_{layeri}}{\sum_{i}^{n} FoliageArea_{layeri}}$$
 Eq. F6

j: day or hour; m: month; i: canopy layer; n: number of canopy layers

Calculated hourly microclimate temperatures are then used to compute corrected local vapor pressure deficits (VPD) used in PHOREAU transpiration computations. These temperatures are also used in GDD calculations (see Eq. 29), as well as for seedling establishment constraints based on minimal temperatures ( $W_{Tmin}$ ). For seedlings, soil-level microclimate temperature is directly used; for established trees, the microclimate temperature is calculated the weighted average height of their foliage area distribution.

Because leaf unfolding and senescence dates are integrated in the calculations of LAI and VCI, the slope of microclimate buffering or amplification can change throughout the year.

While this approach presents a number of advantages, it comes with major simplifications. The most important one is certainly the linear interpolation of microclimate over the height of the stand, which neglects actual wind movement and radiation attenuation dynamics. Microclimatic data, measured at different heights below the canopy, would be needed to calibrate a more realistic non-linear function. Other simplifications include disregarding the effect of soil moisture, ignoring horizontal heterogeneity within patches, and assuming monthly mean temperatures are a good indicator of equilibrium.

Figure F1 | Schematic representation of the slope and equilibrium microclimate approach, reprinted from Gril, Laslier, et al., (2023).

#### **Appendix G: Treewise aggregation**






Because the runtime of a SurEau simulation is driven by the number of distinct water-holding compartments — the atmosphere, soil layers, and mostly importantly tree organs — the first step to reducing the runtime of a SurEau simulation is to reduce the number of initial trees. This approach requires that the global stem volumes and foliage areas remain the same at the stand level, as these are the main drivers of water-use in SurEau and *in natura* (Wullschleger et al., 1998). The aggregation method ensures this through by summing and averaging, at the cost of some precision in the description of the competition for water.

The degree of simplification is specified at the start of the PHOREAU simulation by choosing a number of *classes*: this is the maximum number of aggregate trees created per species at the start for each SurEau run-year. It follows that, for example, a three-class aggregation in a stand with 4 species will result in SurEau initializing with at most 12 trees, which is a more manageable number. To preserve the overall structure of the stand, trees are distributed within classes on the basis of trunk diameter: for an n-class aggregation, for each species, the range of diameters between 7.5 cm and the largest diameter at breast height is decomposed between n-1 segments of same size: classes are then created by grouping all the trees with a diameter at breast height located between the extremities of a given segment, and the last class is composed of all the juvenile trees smaller than 7.5 cm. A consequence of

this method is that a class may contain no tree for a given year, and that trees may move between classes from one year to the next as they grow in size.

After the distribution, a single aggregated tree is created for each class. The volume of this aggregate tree is the sum of the volumes of all the trees in the class; its height the average of their heights; its foliage area the sum of their foliage areas; its root depth the average of their root depths; its root biomass the sum of their root biomasses; and finally its light availability the average of their light availabilities. See Figure W2 for an example case.

#### Appendix H: Dry-year selection








The second optional way of optimizing PHOREAU performance revolves around modifying the rate at which SurEau is called from ForCEEPS. By default, the two submodels are run on a 1-to-1 basis, with SurEau being called at the beginning of each year; but a more parsimonious approach is to run SurEau only for the driest years of the simulation. This simplification is based on the idea that the impact of drought on forested stands, and especially on tree mortality, does not follow a linear curve, but rather depends on climate extremes, physiological thresholds and tipping points (Hartmann et al., 2018). Because this approach requires a prerequisite ranking of all of the years of the simulation according to their dryness, we use an integrative Drought Index calculated for each year (Morin et al., 2021). The rate of SurEau calls — every two years, five years, etc., — is set by the user before the start of the simulation, with a trade-off between runtime and the accuracy of drought-response predictions. At the start of the simulation, the driest year among the first n years is selected as the year SurEau will be called; then, at the start of the n+1 year, the driest year among the next n years is selected, and so on.

# Appendix I: Drought feedback on growth

In assessing the effects of drought events on trees, PHOREAU distinguishes between short-term adaptations and long-term non-reversible consequences — respectively feedbacks on growth and on mortality. The independence of these two mechanisms is key to avoiding confusion between two sources of mortality: that caused by long-term carbon starvation — represented in PHOREAU by diameter growth falling under a certain threshold — and that caused directly by extreme drought through high level of xylem embolism leading to hydraulic failure (Cochard et al., 2021b). A tree subjected to consecutive years of water stress may maintain its conductive vessels but die off due to a lack of carbon intake and defoliation; another may die following a single month of acute water stress despite strong carbon reserves. By establishing a clear distinction between these two pathways, PHOREAU is able to account for the different drought response strategies observed among species.

In PHOREAU, the impact of drought on growth is assessed using the degree of stomatal closure, converted into a drought index *DrI*. Compared to the original ForCEEPS formulation which uses a simple monthly water budget (Bugmann and Solomon, 2000), this new mechanism takes advantage of the detailed hydraulic framework of SurEAU to account for competition for water as well as inter-specific differences in dealing with water-stress. For seedling establishment — for which SurEAU cannot be used — the original drought index *DrI* remains used as a proxy for global stand water availability.

Schematically, as soil water reserves become depleted and soil water potential decreases, trees adapt their conductance by closing off stomata in order to reduce water loss and maintain twig and leaf potentials above cavitation thresholds (Cochard et al., 1996, 2002). This regulation mechanism prevents the premature death of branches and trees due to uncontrolled embolisms, as trees reduce their water loss until only cuticular transpiration remains. The relation between leaf water potential and stomatal closure is an important trait describing a species' response to drought: constrained by a trade-off between carbon gain and risk of hydraulic failure (Brodribb et al., 2003; Venturas et al., 2018), it is correlated with the more often measured turgor loss point (*TLP*) (Brodribb and Holbrook, 2003). While the link between turgor loss and reduced growth is well-documented (Cabon et al., 2019; Peters et al., 2020; Potkay et al., 2022), for PHOREAU stomatal aperture was selected as a continuous variable allowing for a finer feedback.







1700

1705

Stomatal aperture  $\gamma$  in PHOREAU is derived at each time-step from leaf water potential  $P_{L,sym}$  using a sigmoid curve described by two species-specific traits:  $P_{gs12}$  the water potential causing 12% stomatal closure, and  $P_{gs88}$  the water potential causing 88% stomatal closure (Cochard et al., 2021b). Actual stomatal conductance is then calculated as the product between this stomatal aperture ratio and a maximal stomatal conductance value for a given climate. To calculate the drought reduction index DrI of a given tree, daily stomatal apertures ratios  $\gamma_j$  are averaged over the photosynthetic period, which are then averaged over the year (Eq. 19).

#### Appendix J: Drought feedback on defoliation

Between the normal closing and opening of stomata to regulate water flow, and the runaway embolisms responsible for tree mortality after prolonged extreme droughts, trees exhibit a range of intermediate responses to water stress. Among these regulatory mechanisms, the adaptation of leaf area to moderate water stress is of particular importance for any model, such as PHOREAU, which integrates tree growth and drought-resistance.

Water limitation impacts leaf area through three main pathways: the premature shedding of leaves, the disruption of new bud formation (Bréda et al., 2006), and plastic biomass allocation to leaves (Martínez-Vilalta et al., 2004). These mechanisms function at gradually longer time-frames: a cohort of trees may shed their leaves one year in response to extreme drought, and recover their full canopy the next; another may experience several years of decreased leaf area while its leaf phenology cycle is disturbed; and yet another cohort may have permanently shifted to produce less leaf area to adapt to chronic soil water limitations (Limousin et al., 2012; Martin-StPaul et al., 2013). This graduated temporal response is complicated by the fact it is differentially applied among species, following the classic split between drought-avoidance and drought-resistance strategies: indeed, there is evidence that while the reduction of leaf area improves resistance to moderate drought events, it may not avail against severe water stress (Limousin et al., 2022). Furthermore, the short-term gain in drought-resistance of a reduced photosynthetic surface may eventually offset by the negatives consequences of reduced carbon uptake (Poyatos et al., 2013), and the link between leaf area and a reduction of fine root biomass (Gieger and Thomas, 2002).

While the integration of defoliation has been shown to improve the predictions of tree mortality models (Dobbertin and Brang, 2001), this integration is complicated by the fact that few are able to account for the dual role of leaves in carbon-assimilation and water-use. However, unlike most mortality models, the PHOREAU model has the

major advantage of being able to disentangle the contradictory effects of leaf area on growth and drought resistance, and of having an explicit representation of the root compartment with water uptake driven by fine roots and ultimately leaf area (see Sect. 2.4).

#### Appendix K: Drought feedback on mortality









Drought-induced mortality in PHOREAU is derived from the percentage of cavitation, i.e. the percentage of loss of conductance (PLC). This mortality mechanism is entirely distinct from the pre-existing slow-growth mortality in ForCEEPS, and the previously described drought feedback on growth. Indeed, contrary to the slow-growth mortality that reflects carbon starvation and the long-term integrative effects of dehydration coupled with temperatures and competition for light on the capacity of trees to grow and survive (Bugmann and Solomon, 2000), this feedback is only intended to capture catastrophic water failure caused by extreme drought events, irrespective of the overall prior health of the tree. Unlike the stomatal closure used in drought feedback on growth, the cavitation of a tree's hydraulic system is neither quickly reversible, nor does it follow a linear response to hydraulic stress. Furthermore, it occurs only after the stomata have been closed, when, under extreme stress conditions, residual water flow through the cuticle empties the plant's water reservoirs. As water is drained from the soil and the water potential of the system becomes more and more negative, the conductance of a tree's hydraulic system may remain stable until a certain point is reached, when it rapidly decreases as the xylem vessels are embolized and air are formed (Tyree and Sperry, 1989). This non-linear, tipping point response of conductance loss to decreasing water potentials is described by the vulnerability curve of the species. This curve, in the shape of an inverse sigmoid function, is described for each species using a  $P_{50}$  parameter. This parameter, responsible for the main differences in drought-resistance between species (Delzon and Cochard, 2014), is the water potential causing 50% cavitation in the xylem (Cochard et al., 2021b).

#### Appendix L: The rain interception module

Capitalizing on the capacity of PHOREAU to predict individual-tree daily foliage area values that integrate allometry, competition, frost, phenology, and drought-defoliation effects, we implement a rain interception module that reduces incoming rain based on the daily leaf area of the stand. Modelling rainfall interception — defined as free water that evaporates from the leaves and barks of trees after a rain event — is an important component for any model trying to water cycles and tree water balance (Davi et al., 2005a; Granier et al., 1999). The intensity of the interception has been shown to grow linearly with leaf area, for values ranging from 20% to 35% of cumulated rainfall in temperate and continental climates (Bréda et al., 2006). While secondary factors such as irradiance, windspeed, and vapor pressure deficit impact the rate of interception *in natura*, as a first approach we have chosen a simple implementation, inspired from Medfate (De Cáceres et al., 2023b), based solely on daily leaf area, rain volume, and potential evapotranspiration.

A canopy storage volume is derived from the foliage area of the stand. This volume is incremented at a daily timestep with incoming rainfall, and outgoing evaporated water. For a given volume of incoming rainfall, the throughfall, or the volume of water to reach the ground, is calculated with a simplified *Beer-Lambert* formula, in a similar fashion to the way light extinction is computed. Because the canopy storage volume is itself limited, any intercepted water that overflows this maximal quantity flows down the soil; a natural consequence of this property is that a given volume of given rainfall will yield a greater cumulated throughfall when concentrated in a single day, than when distributed over several days with intervening evaporation. The algorithm, presented below in Eq. L1, computes the daily stand-wide throughfall volumes that then serve as inputs to the water balance model.

```
CanopyStorageCapacity_{j} = \frac{LAI_{j}}{2}
PotInterceptedRainfall_{j} = Rainfall_{j} * (1 - e^{-0.5 * LAI_{j}}) \qquad Eq. LI
PotThoughfall_{j} = Rainfall_{j} * (e^{-0.5 * LAI_{j}})
CanopyStorage_{j} = CanopyStorageCapacity_{j} - CanopyStock_{j-1}
Throughfall_{j} = \begin{cases} PotThoughfall_{j} & PotInterceptedRainfall_{j} \leq AvCanopyStorage_{j} \\ Rainfall_{j} - AvCanopyStorage_{j} \end{cases} PotInterceptedRainfall_{j} > AvCanopyStorage_{j}
StoredWater_{j} = Rainfall_{j} - Throughfall_{j}
CanopyStock_{j} = Max(0, CanopyStock_{j-1} + StoredWater_{j} - PET_{j})
j : day of year
```

## Appendix M: The bootstrap algorithm




In the PHOREAU framework, the leaf area is updated at the end of the year, after each tree's crown length has been updated according to the light availability. However, the light availability that is used to calculate the new crown lengths is the result of the stand area of the previous year, which is itself the result of the previous year's crown lengths. This asynchronicity means that — disregarding other processes like growth regeneration and mortality — the estimation of stand area will oscillate around an equilibrium state. While this equilibrium state is dynamically stable, the oscillations for the first few years are large enough to be significant. This is especially problematic when starting the model from an inventory: because actual crown lengths are rarely available, the model is forced to initiate the crown at the maximum species' value; the resulting very low light availability means that the following year the crown lengths will be reduced by a large factor, which means that more light will be available the year after that, causing a new spike in stand leaf area. It is to correct for this effect that we implemented a bootstrap algorithm where, before the first year of the simulation, multiple iterations of the light competition module are run until the shift in stand area between two successive iterations becomes negligible

**Figure M1** | **Illustration of PHOREAU canopy bootstrap algorithm.** Top: one-sided leaf area indices predicted by the PHOREAU bootstrap algorithm, initialized with a Picea abies dominated inventory (RENECOFOR EPC 39a, 2003). Bottom: three snapshots of predicted foliage area and light availability vertical stratification at different steps in the algorithm. For details on the calculation of the Vertical Complexity Index (VCI), refer to Appendix F.

# Appendix N: The Integrated Carbon Observation System

The Integrated Carbon Observation System is a network of stations that measure ecosystem-atmosphere exchanges of greenhouse gases and energy at a high frequency (Baldocchi, 2003), using the eddy-covariance technique. In addition, a large set of ancillary variables needed for the interpretation of the flux data are also measured: for forest stations these include, among other measures, tree inventories, leaf area index, and soil data — all of which can be leveraged for our modelling purposes (Gielen et al., 2018). The large scope of measured variables in ICOS framework makes any validation based on it easily scalable, and will in the future allow testing of any newly integrated PHOREAU processes (such as carbon retention or vertical micro-climate interpolation). Finally, a set of rigorous specifications for the installation of the eddy-covariance tower sensors, and a common pipeline for the post-processing of the raw data through the Ecosystem Thematic Centre (ETC), ensure the high level of comparability between sites that is necessary for large-scale model evaluation.

# Appendix O: Puéchabon









The Puéchabon experimental site (43°44'30"N, 3°35'40"E, altitude 270 m) is located in a forest of holm oak located in the South of France near Montpellier. With its last clear cut in 1942, and managed as a coppice for centuries before that, the site is characterized by a high density (5000-7000 trees/ha) of small (5.5 meter high overstorey) *Quercus ilex* trees: they make up an old forest with a basal area of 30 m²/ha, (Rambal et al., 2014), and an LAI around 2.2 with little seasonal variability. Located on a flat area, with a rocky soil of Jurassic limestone filled with clay, its small water reserve (roughly 130 mm of water over the 5 meter profile) and typically Mediterranean precipitation pattern (highly variable from year-to-year, with a measured range of 550 to 1550 mm primarily concentrated between September and April) made it an ideal candidate to study the long-term effects of drought.

Within the framework of the *Mediterranean Terrestrial Ecosystems and Increasing Drought* (MIND) project, the diameter of trees contained in twelve 100m<sup>2</sup> plots have been measured on a year-to-year basis since 2003: these are distributed between three control plots, three thinned plots (33% reduction of basal area), three plots with partial rainfall exclusion (33% throughfall), and three thinned and rainfall excluded plots (Gavinet et al., 2019b). We have used these plots to run simulations from 2003 to 2020, and assess how the PHOREAU model simulates the effects of tree density on drought resistance.

## **Appendix P: Font Blanche**

The Font Blanche experimental site (5°40'45''E, 43°14'27''N, altitude: 420 m) is located in a mixed-forest of Aleppo pine and holm oak, with an overstorey of *Pinus halepensis* (13.5 m height) that dominates a coppice of *Quercus ilex* (6.5 m height). With a basal area of 21.3 m²/ha and and LAI ranging between 2.5 and 2.7 it is less dense than Puéchabon, but otherwise boasts a broadly similar soil and meteorological profile (Simioni et al., 2016). For our validation we used the 625m² control plot (PM30) of the rainfall exclusion experiment, in addition to the main plot of 6400m² that we split between 25 smaller splots of 267 m² apiece to satisfy PHOREAU homogenous competition assumptions. Our timeframe for this site ranges from 2007 to 2020.

# **Appendix Q: Hesse**






The Hesse Experimental site (7°3'59''E, 48°40'30''N, altitude: 300 m) is located in a beech (*Fagus sylvatica*) stand in north-eastern France, on a plain at the feet of the Vosges mountains. Average tree height was measured at 16.2 m in 2005, with a maximum leaf area index over 7.5 indicating a very high level of canopy closure. In comparison to the two previous sites it is characterized by a wetter, semi-continental temperate climate, with a deep loam-clay soil (Davi et al., 2005a; Dufrene et al., 2005). Unlike most sites in the ICOS network it is fertile, fast-growing and subjected to frequent thinnings, with an average tree age of only roughly 40 years in 2005, allowing us to test the capability of PHOREAU to simulate canopy and basal area regrowth after a cut. Furthermore, despite the stand having high rainfall and soil high water holding capacity, droughts events are responsible for most of the interannual variability in tree growth (Granier et al., 2008). We extracted from the inventory four evenly sized 300 m² plots. Because the validation timeframe ranges from 1999 to 2010 when the most data was available (Cuntz et al., 2023e, 2023d, 2023c, 2023b, 2023a; Betsch et al., 2011; Peiffer et al., 2014; Tuzet et al., 2017; Zapater, 2018), our model also replicates two thinnings that occurred in 2005 and 2010, respectively for 25 and 15% of the basal area.

#### Appendix R: Barbeau

The Barbeau experimental site (2°46'E, 48°28'N, altitude: 100 m) is located in the Fontainebleau national forest southwest of Paris. The stand is dominated by sessile oak (*Quercus petraea*) trees that 25 m at 100 years of age, with an understory of hornbeam (*Carpinus betulus*). Mean annual cumulated precipitations of 677 mm are evenly distributed over the year, and feed into a deep soil with roots able to reach at least 150 cm in depth. We initialized our validation over 9 plots of 1000 m² using an exhaustive inventory made in the winter of 2006-2007; we ran running it until 2021, including a thinning in 2011 (Delpierre et al., 2016; Maysonnave et al., 2022). Unlike the other studied sites, growth data was not available on a tree by tree basis, but instead aggregated at the stand level (Briere et al., 2021).




# **Appendix S: Supplementary Tables**

Tables S1 to S17 are available in the supplements published alongside this article.

#### 1900 Appendix T: Climate Reconstruction

The SILVAE web portal (Bertrand et al., 2011 and Richard, 2011) offers monthly average temperature and precipitation sum data over France at a finer spatial resolution, accounting for microclimatic differences caused by differences in altitude, exposition, and wind orientation. These time-series, available for the period between 2000 and 2014, were used to correct the coarser ERA-5 Land dataset for all variables except wind-speed: either by direct mean-adjustment for the average temperature and precipitation variables, or after a prior linear regression of the variable over the mean temperature for the given month of the ERA-5 Land time-series. For the average temperature variable, between 2000 and 2014, daily values were corrected by the difference between the average of all the daily ERA-5 Land values for that month and the single monthly value of the SILVAE correction dataset; whereas for the years outside of this range where the corresponding monthly value was not available, the difference

was calculated using the mean of all values for the given month between 2000 and 2014. A similar method was used for the precipitation variable, where the daily values were multiplied by the ratio between summed monthly ERA-5 precipitations, and the single monthly SILVAE value. For the other variables except wind speed the same method was used as for the average daily temperature, except the addition factor was itself first multiplied by the slope of the regression between the temperature and the variable. The wind speed variable was not corrected due to its weak correlation to mean temperature. The workflow for climate reconstruction is summarized in Figure 7.

### Appendix U: Evaluation against leaf area









The importance given to competition for light and leaf area prediction is one of the core principles of ForCEEPS — and the FORCLIM and FORECE models before it. However, because the initial models were focused on long-term forest dynamics, the methodology used to calibrate and validate the light competition module was based on a broad adequation between expected LAI values, and those reconstructed by the model after runs of hundreds or thousands of years starting from the bare ground (Kienast, 1987). Even then, LAI was not usually considered in the final validation, which was made on predicted biomass, basal area, tree density, or species composition (Bugmann, 1996; Wehrli et al., 2006). Notwithstanding the fact that this approach disregards past human interventions in the observed stands, it only accounts for equilibrium states, which becomes problematic when one wishes to apply the model at shorter timescales and consider the shorter-term effects of climate-change on existing forests. Yet, while ForCEEPS did use actual inventories and short-term productivity for its original evaluation (Morin et al., 2021), its performance was not assessed by comparing simulated and observed predicted leaf area index values.

This approach holds up as long as leaf area can be considered to be an intermediary variable. Because the previous models only used leaf area within the framework of their light competition modules, a given tree's predicted leaf area only mattered insomuch as it provided shadow to neighboring smaller trees, decreasing their light availability factor. In this respect, absolute leaf area mattered less than the relative distribution between trees and species, which governed growth and final predicted compositions.

However, in PHOREAU, tree leaf area is also an integral input of another part of the model: the simulation of hydraulic processes. This is because the upwards flow of water through the tree is ultimately driven by the transpiration in the leaves (Ruffault et al., 2022). And, in this respect, water flow is driven not by the relative, but by the absolute quantity of leaf area. Mechanically, a stand with a greater total leaf area index will tend to exhaust its water reserves faster; and tree leaf area, in ecosystems subjected to drought, is directly modulated by recent drought events (Bréda et al., 2006). These mechanisms, which are implemented in PHOREAU, require an accurate prediction of yearly stand leaf area index as a prerequisite condition to any simulation of hydraulic stress.

Unlike other validations of SurEau (Ruffault et al., 2023), the PHOREAU framework prevents the direct use of leaf area index as an input to the model; instead, the model initializes the stand LAI using solely the diameter and height information contained in the initial inventory. This makes the model suited to work on a majority of sites, where trunk diameters are measured but not leaf area, and allows it to make predictions in the future, as the LAI is recalculated on a year-to-year basis. The drawback of this approach is the addition of a new source of error when

LAI is wrongly estimated. This is why, before validating the model on growth or drought-induced mortality, a preliminary validation of the leaf area index predictions was necessary.

# Appendix V: Height-Diameter Interpolation






Height-diameter ratio interpolation. In order to leverage PHOREAU's ability to reproduce stand light availability

and microclimatic conditions based on the structure of modelled trees, we used the newly independent tree height variable (see Sect. 2.1.2) as an input parameter. However, height measurements were only available for a subset of trees across all RENECOFR and ICP II plots. Therefore, for trees where only circumference was measured, we applied plot-specific LOESS local regressions (Cleveland and Loader, 1996) to estimate species height-to-diameter curves from available measurements. The variability in height-to-diameter relationships among plots can be seen in Fig. U1 and Fig. W16,

Figure U1 | Diversity of site height-to-diameter curves for *Fagus sylvatica*. Refer to Table S3 for details.

contrasted with the fixed height-to-diameter formula used in the original ForCEEPS framework. The associated statistics presented in Table S3 highlight the general tendency of the formula to underestimate tree heights in our study sites (AB = -15.7%; Table S3); this is not necessarily surprising, as the RENECOFOR and ICP II sites mostly support denser, more productive stands, where trees prioritize height growth to compete for sunlight.

# Appendix W: Supplementary evaluation figures

**Figure W1** | **Predicted distribution of stand leaf area and light availability.** This figure illustrates the vertical gradient of predicted light availability indices of the four considered ICOS sites for specific simulation years. The light availability is presented over the aboveground profile, divided into 0.1 m layers. In addition, the area of each shape in the layers represents the predicted aggregate leaf area. Refer to Fig. W17 for light availability index gradient. The figure also includes global annual stand parameters LAI and VCI (see Appendix F of VCI).

**Figure W2** | **Predicted fine root area distribution over the soil profile.** For the four ICOS validation sites, for certain simulation years a partial vertical soil profile is shown, with the overall dryness of each soil layer depicted as a gradient using its 10<sup>th</sup> quantile relative extractable water (REW) percentage. For each species and size class aggregate tree (refer to Appendix G for details on the aggregation method), the distribution of the inverse cone along the soil layers represents the predicted location of its fine roots, with its total aggregate fine root area index (FRAI) shown under. Refer to Fig. 8 for species and cohort color codes.

**Figure W3** | **Predicted versus observed annual stand leaf area index (LAI).** For each simulation site, the bars depict the annual leaf area index projections generated by the PHOREAU model, broken down by species and size class contributions (refer to Table S16 for associated statistics). The dashed line represents the observed annual stand leaf area index (data sources are detailed in Table 1). Leaf area index is defined as the total one-sided leaf area per unit of ground area. Refer to Fig. 8 for species and cohort color codes.

Figure W4 | Predicted versus observed evolution annual stand basal area loss due to mortality. For each simulation site, the bars depict the summed annual total basal area ( $m^2$ /ha) of all dead trees, broken down by species and size class). Observed values are derived from stand inventories, while predicted values are generated by the PHOREAU model. Also shown are the yearly basal area loss rates, calculated relative to the initial basal area for two distinct time periods in each simulation, along with the total basal area dieback per hectare ( $G_{dead}$ ). Transparent bars indicate years with thinnings (see Appendices Q and R for details), which are excluded from the mortality statistics. Refer to Fig. 8 for species and cohort color codes.

Figure W5 | Predicted versus observed daily real evapotranspiration (ETR) and Transpiration. For each simulation site, the plain blue line is the regression line of the linear model of the relationship between observed and predicted stand daily ETR (or transpiration for Hesse), with confidence interval represented with the grey dashed lines; the dashed red line is the 1:1 line. See Table S11 associated statistics. Color code for the seasons as follows: ,Winter; ,Spring; ,Summer; ,Autumn

Figure W6 | Evolution of predicted versus observed stem water potentials. For the dominant species of the four ICOS simulations, the blue line depicts the daily evolution of the stem water potentials (MPa) generated by the PHOREAU model and averaged over the aggregate trees of the species (refer to Appendix G for details on the aggregation method). The red points represent the observed water potentials, limited to the years for which observational data is available (data sources are detailed in Table 1, and associated statistics in Table S10). For Puéchabon, Font Blanche and Barbeau sites, the minimum daily observed and predicted water potentials are shown. For Hesse, where only predawn observations are available, the