# Peer review of "PHOREAU v1.0: a new process-based model to predict forest functioning, from tree ecophysiology to forest dynamics and biogeography"

_EGUsphere, 2025_

## Author Comment (AC1)

*We would like to thank Referee #1 for the interest taken in our study and the helpful suggested points of improvement. Attached is a pdf with our answers in red italic type for easier readability. Unfortunately, we can not attach the revised manuscript here, making tracking line changes a bit difficult; but if requested we can send a version with the correct line numbers, and red type for modified sections.*

**General Comments**

A minor suggestion would be to more frequently remind the reader in the technical sections (e.g., 2.1.2) that additional details are available in the appendices. Given the manuscript's length, this would help reader navigation. Similarly, a clearer, earlier statement regarding the different time-steps (e.g., daily, hourly) and the multi-layered soil structure used in the water modeling would help orient the reader from the outset.

*Reply: A statement summarizing these aspects of the model has been added at the beginning of section 2 (L202-205). Additional references to the appendices have also been added in the relevant sections.*

The presentation of the evaluation results could, however, be improved. In several figures, font sizes are quite small, and some plots appear stretched. For long time series of daily data (e.g., Fig. 10), the fine temporal scale is lost, making them difficult to interpret. Presenting more of this data as scatter plots (as in Fig. 13) could enhance clarity. To improve readability and focus given the paper's length, the authors might also consider moving some detailed results to the supplementary materials, while retaining the key findings from each evaluation level in the main text.

*Reply: We have moved figures 12 and 15 to appendices to reduce the paper's length while focusing on the most important and readable results. Regarding the time-series in Figure 10 and 11; while it is indeed difficult to finely interpret the variations between days or even two months, they do make visible the variations between summer and winter months, and particularly whether the model is able to capture ETR peaks and soil draining in summer; furthermore, they allow us to make distinctions between simulation results from earlier and later years, which as important as deviations in stand structure can accumulate over time (e.g. Hesse).*

**Specific Comments**

**Introduction:**

- The introduction provides a very detailed account of species mixture effects (e.g., lines 70-95), which could be shortened. On the other hand, a broader context of other forest modeling approaches is a bit lacking.

*Reply: We have attempted to remedy this by inserting, near the end of the introduction (L161-174), a paragraph that zooms out and replaces the development of PHOREAU within the broader context of 'vegetation dynamic models'.*

- **L105-110:** The text mentions "identified two main shortcomings in forest models." It would be helpful to briefly elaborate on *how* these specific shortcomings were identified (e.g., through literature review, previous modeling experiments, etc.).:
  *Reply: These weaknesses in predicting tree mortality and regeneration were identified through a review of litterature on gap models, as well as it has been highlighted in former studies (Bugmann and Seidl 2022 – cited in the manuscript): these processes were rarely evaluated directly, but rather integratively with predicted species distributions and site basal areas. The paragraph has been amended to reflect this (L106-109).*

**Model Description:**

- **L159:** Typo: "plaform" should be "platform". *Amended*

- **L216 (Eq. 1):** There appears to be a layout issue in the equation. It should likely read 2*H_max,s - b_s * e^(...) rather than having the allometric parameters in the denominator's exponent. Please verify the formula. *Amended*

- **L232:** Suggest inserting the word "species": "shade intolerant *species* having...". *Amended*.

- **L251-265:** The concept of "crown ratio reversion" needs clarification. Does this mechanism allow the base of the living crown to move downwards again, effectively re-greening parts of the stem that were previously bare? This should be clarified here and in the appendix. *Yes this is the intent of the new mechanism. The paragraph has been clarified (L312-314).*

- **L277:** The symbol for the clumping factor appears to be missing from the parentheses: clumping factor ( ). *This is very strange, the $\Omega$ symbol is visible in the uploaded pdf on our end.*

- **L333:** While "symplasm" is defined by contrast, a brief explanation of "apoplasm" (the continuum of cell walls and extracellular spaces) would be beneficial for non-specialists. *Description has been added (L395-397).*

- **L368:** Typo: "depending on depends". *Amended*.

- **L382:** It would be useful to briefly explain what a "semi-implicit solver" is and give an indication of the runtime difference it makes compared to an explicit solver. *Some*

*further details have been added in the text (L446-449); but readers should refer to the Ruffault 2022 paper for a comprehensive description of the pros and cons of each method.*

- **L531 (Eq. 13):** There may be a typo in the denominator. The text reads P_gSBB, which seems inconsistent with the parameter P_gs88 defined on L525. Please check. *Amended.*

- **L545:** The growth reduction factor GR_crown is used here but has not been fully explained. A brief definition is needed. *Explanation has been expanded (L605-608).*

- **L553:** The citation Hammond et al., 2019 appears to be missing a closing parenthesis. *Corrected typo.*

- **L620:** The rationale for weighting light availability by daily mean temperature needs more justification. This method heavily weights hot summer months when growth may be limited by other factors (like drought). A temperature response curve that is primarily limiting at the cool end of the spectrum might be more ecologically realistic.
  Reply: *This explanation was indeed misleading: it is in fact not average temperature but rather GDD that is used to weight light availability. Furthermore the disregarding of drought-stress in the formulation is a result of the fact Phenofit was linked with ForCEEPS before SurEau : further developments should indeed take full advantage of drought-stress prediction. This limitation was aknowledged in the test (L681-684).*

- **L651 (Eq. 19):** Notation for leaf unfolding and coloration intervals is slightly inconsistent between the text (Uls, Cls) and the key below the equation (UI_s, CI_s).
  Reply: *It seems that this is a formatting issue from the GMD site because we have exactly the same notations on our end (now Eq. 24).*

- **L695:** A word appears to be missing in "the fine root area of a tree in a determines...". *Amended.*

- **L749:** There is an extraneous character (a hyphen) after the period at the end of the paragraph. *Amended.*

**Results, Discussion & Figures:**

- **L765/Figure 4:** This figure is difficult to interpret due to very small font sizes and hard-to-distinguish color-coding. Furthermore, the claim that it shows "acclimatization" over 1500 years seems more likely to reflect changes in stand structure and species composition rather than plastic adaptation within individual long-living trees. Please clarify.
  Reply: *Due to concerns about the readability of the figure and its overall quality, we have decided to move it to supplementary information (W17). It is indeed the acclimatization*

*effect and not structural changes that are responsible for the overperformance of simulation B (with a moderate drought event before the main drought event) compared to simulation A, and we had confirmed this by looking at simulation results obtained without the new root plasticity module, where simulation B does not overperform simulation A. Unfortunately this is not apparent in the figure, and we have updated the caption to reflect this.*

- **L880/Figure 6:** There appears to be an inconsistency in the visualization. For instance, for the Puéchabon site, the caption states "3 patches of 100m²", but the grey grid lines on the ground seem to depict a different arrangement (e.g., 4x4 grid). This should be checked for all subplots.
  *Reply: Indeed gridlines are a purely visual artifact of the Capsis vizualisation, the caption has been amended to reflect this.*

- **L914:** The paper states that longer time-lapses "would have mechanically improved simulation results". This is counter-intuitive, as one might expect simulating longer periods to be more challenging and prone to error accumulation. Could the authors please clarify what is meant by "mechanically improved" results in this context?
  *Reply: This statement has been reworked and nuanced (L922-926). In fact, when looking at simulations results, the length of the simulation had no significant impact on prediction error (Fig. W11e). By longer time-lapses we meant that focusing on yearly patterns is a harsher test (and thus more relevant) than doing so on decades or more. In fact, while longer simulations are indeed more prone to error accumulations, they are also more forgiving to errors in predicting single-year deviations from the norm caused by extreme climatic events. It is this second effect which we are more keen to evaluate for PHOREAU, in view of its application to predicting the effects of climate change. Of course, the best situation would be to use long time-series with fine temporal resolution as reference to evaluate a model's predictions. Yet, such data are still scarce for trees physiology, except in some key sites such as the ICOS ones.*

- **L1019:** Typo: "crown Al ratio" should likely be "crown ratio" or similar. *Corrected error*

- **L1059:** There is a minor date discrepancy for the Hesse site thinning. The main text mentions a cut in 2005, whereas Appendix Q lists thinnings in 2004 and 2009. This could be harmonized for clarity. *This has been amended.*

- **L1250/Figure 17:** This figure effectively illustrates the model's performance across ecological gradients. A very nice visualization.
  Reply: We thank the reviewer for this statement!

- **L1375 (and elsewhere):** The citation 'Allen, Macalady, Chenchouni...' is very long. This format occurs multiple times (e.g., L54) and could be consistently shortened to 'Allen et al.' for readability. *Yes this was very strange! Amended.*

**References:**

- **L2231:** The reference for Bréda, Soudan and Bergonzini is listed with "(no date)", which is unusual and could be clarified. *Amended.*

---

## Author Comment (AC2)

*We would like to thank Referee #2 for the very thorough review and the many suggested points of improvements. Attached is a pdf with our answers in red italic type for easier readability. Unfortunately, we can not attach the revised manuscript here, making tracking line changes a bit difficult; but if requested we can send a version with the correct line numbers, and red type for modified sections.*

**General comments**

Section 1, Introduction: I think the introduction is for the most part concise and accurate. My main suggestion is to try to present PHOREAU in a broader context of vegetation demography models. A few references that could be interesting as starting points are Fisher et al. (2018) (https://doi.org/10.1111/gcb.13910) and Bugmann and Seidl (2022) (https://doi.org/10.1111/1365-2745.13989).

Reply: *We are grateful for the suggestion and references. We have tried, near the end of the introduction (L161-174) to briefly zoom out and replace the development of PHOREAU within the broader context of VDMs.*

Section 2, Presentation of the model: I think this section could benefit from some extensive restructuring. Currently, the sub-sections have several back and forth points (e.g., section 2.1.1 points to specific equations in section 2.4.5), and in many cases equations are shown in blocks (e.g., near lines 425-440) that do not really describe each and every term. In addition, some information is presented in sub-sections with titles that do not describe them (e.g., rain interception is presented in section 2.4.3 "leveraging leaf phenology and hydraulics to temporalize competition for light"). Part of this may have stemmed from the authors seeking to keep the sub-sections aligned with the contributing models (ForCEEPS, PHENOFIT, SurEAU). Whilst I appreciate this, I think it is more important to streamline the description of PHOREAU and organise sections according to processes, and refer to the original models for the specific modules as needed. I also think it would be better to start with a brief overview of the model (as the authors already do), describe the initial and boundary conditions needed by PHOREAU, present the fundamental equations in the first subsection (i.e., combine 2.1.1 and 2.4.5), then split the other sections by process. To link specific processes to each originating models, the authors could consider adding a table in section 2 that lists all processes, their time scales, the originating model and the sub-section where they are described. Finally, it would make it much easier to follow this section if the authors presented the equations and terms in the same order as they first mention them.

Reply: *The organization of the model presentation section was indeed a point of discussion among the co-authors, as we tried to strike balance the two extremes of a full presentation of the PHOREAU model without reference to the parent models (which would have considerably lengthened the manuscript), and a presentation focusing only on new developments (with the*

*risk of obscuring general model functioning). We agree that we have strayed too much towards the latter option. To rectify this, and in keeping with your suggestions, we have combined sections 2.1.1 and 2.4.5 by presenting the fundamental demography equations (growth, regeneration and mortality) at the beginning of the model presentation, referencing later sections for details on the calculations of the underlying factors. This avoids much of the backtracking between sections, and equations are now mostly refered in order of appearance. We have consequently amended several section titles to more accurately reflect their content. While ForCEEPS is no longer presented independently from PHOREAU, we have opted to keep separate independent sections for PHENOFIT and SurEAU: this is justified the fact the phenology module in PHOREAU is quite separate from the rest of the model; and our wanting to highlight several developments for SurEAU presented for the first time in this paper. We have also added (Table Z1) a table that list processes with their timescale and originating model.*

Section 2.1.1 and 2.4.5. I missed one equation that brings together establishment, growth and mortality to describe the change in forest characteristics (stem number density or basal area). Even though it may be a bit obvious, it would help visualise how PHOREAU resolves forest dynamics.

*Reply: Unfortunately a global equation of this kind does not exist, as changes in stand structure are the outcome of the independent yearly regeneration, growth and mortality equations, now presented at the beginning of the paper. For instance stem number and stand basal arise from interactions between coexisting individual trees, while the response of each tree depends on its species, age, size and social status (dominated or not, root depth…), as well as environmental conditions.*

Sections 3 and 4. I truly appreciate that the authors acknowledged that PHOREAU is a process-rich model and provided a comprehensive assessment of the model that went beyond a basic comparison of a single metric. I also liked that the authors appraised the model performance under multiple climates across Europe, and pointed out where the model predictions work best and has issues. That said, in part because the paper is rather long, the results of this comparison are only briefly described, and some of the comparisons may need more explanation, because of the multiple assumptions on both the model and the observations. I think that if the authors keep this as a single paper, they should simplify many of the analyses. For example, their Figures 10 and 11 have several simulation details (e.g., individual contributions of transpiration and evaporation from soils, trunks and canopy interception), but section 4.1 mostly describers the comparison of total ET. If the individual components do not help explain differences, the authors could simplify the figures and only show what they can describe. Likewise, Figure 12 is only mentioned as a support to explain Figure 10, and little is said about PHOREAU's ability to

represent the seasonal cycle of stem water potentials, and how the model perfomance varies across the sites. More critically, the comparisons with observed forest inventories use potential vegetation simulations. This may be the comparison that is feasible, but most of the forests are managed, so it is unlikely that the discrepancies are entirely due to PHOREAU being unable to represent forest structure and composition.

*Reply: We thank the reviewer for pointing out the main strengths of this study. In the spirit of this comment, we have striven to reduce the length of our result analysis, and select the graphs we were best able to interpret. For example, productivity evaluation has been concentrated on the most relevant stand growth level, while tree-level productivity validation has been moved to appendices.*

*Concerning the simulation across the ICOS sites, in our attempt to homogenize different results across the four sites we had indeed glossed over some of important differences in data measurement and availability between sites (although these are mostly detailed in the site-specific appendices), especially concerning Hesse upscaled transpiration. We have provided more details on these differences, while moving extraneous results to appendices, notably Figure 12 as the majority of water potential measurements being concentrated in the summer months made it difficult to interpret seasonal patterns. Concerning our choice to distinguish between simulated sources of ETR in Figure 10, while we can only evaluate the aggregate stacked results against flux tower observations, we feel this visual breakdown has value in demonstrating model functioning, explaining for example why Barbeau has residual ETR in winter months, or the difference between the nature of the comparisons for Hesse and the three other sites. Concerning comparisons with observed forest inventories, we think there may have been a misunderstanding (although maybe only on our end!). In Section 4.2, 4.3, and 4.4, we do not use potential vegetation simulations; PHOREAU is initialized on existing exhaustive forest inventories as detailed in Section 3.2.3, preserving exact forest composition and near-exact forest structure (this is notably evaluated in 4.2 for stand leaf area); then for each site we compare predicted to observed productivities over a few years in which we know there to have been no management, disregarding regeneration both in observations and simulations. In Section 4.5, we do use potential natural vegetations as the aim in this part was to assess the model's ability to simulate realistic vegetation composition (and because this was classicaly done to evaluate gap models, see eg. Bugmann 1996 and Morin et al. 2021 – cited in the manuscript). Therefore, the simulations carried out for this section have been initialized on bare soils and run for a long time period (eg. 2000 years), retaining only climatic and soil data from the simulation datasets (these could in truth have been any other points in Europe, we only retained the same 340 points as in earlier sections for the sake of convenience).*

*Finally, regarding the possible split of the paper, this was a hot topic between co-authors. We discussed about the possibility to have two papers, as it would have led to two shorter articles,*

*probably easier to read. Yet, we thought it was better to have all information in a single paper, to avoid back and forth reading when the presentation and evaluation of a model are shown in two separate pieces. Last (but unfortunately, not least), we would like to mention that this study is part of a PhD project, which imposes us some constraints about the publication timing of this work. Splitting this paper in two papers would probably delay the process, which would make things a bit difficult administratively.*

**Specific and minor points**

Figure 1. Considering that PHOREAU is an integration of three models, it would be helpful to indicate which processes in the figure are primarily coming from ForCEEPS (e.g., add ForCEEPS beneath the "competition for light" bubble. *Amended*

Section 2.1.2. I suggest citing each individual appendix near the specific process that they describe in detail. *Amended*

Lines 161–163. The authors refer to the Capsis modelling platform multiple times throughout the text. Even though they provide references, it may be worth describing briefly what this platform is and does, for those readers unfamiliar with the platform.
*Reply: A brief description was included at the beginning of section 2.4.1*

Eq.1. The denominator seems to have some formatting issue. *Amended*

Line 261. This is a bit cryptic. When the authors refer to "crown reversion when light availability increases", are they referring to increased light availability due to changes in forest structure, or something else?
*Reply: We refer to the possibility of previously dominated trees - whose crown has been reduced because of weak light availability, to re-grow part of their crowns when some or all of their competitors disappear (because either death or harvest). The explanation was indeed lacking, and we have expanded it (L315-317).*

Line 285. Is SLA the only variable controlling the diversity of drought resistant strategies in PHOREAU? If so, provide a high-level overview of how this works in the model (e.g., through empirical relationships with X, Y and Z).
*Reply: SLA is only one variable among many that explain differences in plant resistance to stress: other variables, mostly coming from SurEau, are described in table S13 and include the P50 (water potential causing 50% cavitation), stem specific conductivity, and maximal and minimal stomatal conductances. The paragraph has been amended to reflect this (L341).*

Figure 2. Replace "density-dependant" with "density-dependent". *Amended*

Line 314. The correct term seems to be meteorological data, unless PHOREAU expects long-term averages for each day of the year. In addition, no need to change the model, but from reading the paper I got the impression that PHOREAU takes ERA5-Land data aggregated by day, then uses some internal procedure to disaggregate the data to hourly. I suppose in the future it would be better to take hourly data directly from ERA5-Land.

*Reply: We had not considered this, as the original models had been developed based on more limited meteorological datasets with only daily (or even monthly) data. This does warrant further development.*

Lines 321-326. This explanation appears multiple times throughout section 2 (e.g., 454-456, 465-468). I suggest adding this information only once, when providing the model overview early in the section, and dropping all the other instances.

*Reply: We have dropped the redundant explanation in the Phenofit section. However because the SurEau version developped in Capsis is presented here for the first time, we feel it is important to underline that it has several new features absent from previous R versions, specific to Java object-oriented programming.*

Lines 328-334. The authors describe the main characteristics of the symplasm, but not the apoplasm. *Description has been added (L.396-398)*

Line 351. Replace "phenomenons" with "phenomena". *Amended.*

Line 356. There seems to be a problem with the parentheses. *Amended.*

Line 383. Consider dropping the word "exactly", as it implies bit-for-bit comparability, which is unlikely to be the case. *Amended with « nearly indentical ».*

Line 483. Repetitive, consider dropping the sentence. *Sentenced has been dropped, and paragraphs combined.*

Line 501. How are the results disaggregated? Does the model assume that all trees had the same values for the predictors of growth and mortality, or does it assume some sort of distribution? Either way is fine, but a bit more detail would be helpful.

*Reply: All trees of a class have the same drought growth reductor, and the same probability of extreme drought-induced death. An explanation has been added (L.564-568).*

Figure 3. This figure currently has too many symbols and details that are not really described in the figure, the caption or the paper. Perhaps they are explained in the original papers. Additionally, the image quality is a bit poor, so it is hard to read. I suggest replacing this figure with a simplified version of it, where only the key processes are spelt out more clearly.

*Reply: We have replaced this figure with a simplified higher quality version, retaining only key processes.*

Lines 551-554. Does this assumption influence the seasonality of variables that may depend on PLC, such as GPP or ET?

*Reply: Yes, by delaying PLC reparation until the end of year instead of during the growth period, the model likely overestimates the average amount of cavitation throughout the year, which in turns leads us to underestimate yearly evapotranspiration, and yearly growth (through an overestimation of stomatal closure caused by a lack of conductance in the stem). The actual magnitude impact is hard to quantify; and, given the model architecture, could not have been easily avoided.*

Line 565. Consider replacing "random" with "spurious", unless referring specifically to a process that is represented by a random variable in PHOREAU. *Amended, they are indeed not random.*

Lines 619-623. How does the weighted average by mean daily temperature works? Does PHOREAU use absolute temperature (Kelvin) or a relative scale (e.g., Celsius). This would give very different results… Also, if the latter, how does the averaging work when the temperature is at or below freezing?

*Reply: The explanation was indeed lacking and misleading. Actually, the weighted average is done by weighing the daily GDD value against the annual mean value. A day with an average temperature below the reference GDD temperature (set at 5.5 C°) will therefore have a weight of 0. This has been corrected in the text (L678-681).*

Line 679. Unless I missed it, this is the first occurrence of SurEau-Eco, this needs a description.

Reply: SurEau-Ecos is the name for one of the previous SurEau versions coded in R. Because the description of the SurEau model history has been simplified compared to earlier drafts, this reference no longer makes sense, and has been corrected.

Lines 689-690. This part is a bit confusing. What are the inputs the user is supposed to provide?

*Reply: It is the area of each patch that is a simulation input. A simulation with trees split in 5 patches of 1000m2 will differ from another with the same trees but split in 10 patches of 500m2, because competition for light and water only takes place inside the patch. The description has been amended to make this clearer (L767-770).*

Line 705. Consider using the present tense instead of future tense. *Amended.*

Lines 713-716. Are there any assumptions on the vertical distribution of roots within the rooting zone (uniformly distributed, exponential decay).

*Reply: The fine root area is distributed between soil layers using a negative exponential model, as in previous SurEau iterations. An explanation has been added (L778-779)*

Figure 4. The specific species and sizes are a bit hard to read due to the colours being similar to the background. Also, would it be possible to disaggregate the above-ground contributions of each species to the basal area and LAI, similarly to what is done in the below ground.

*Reply: Due to concerns about the readability of the figure and its overall quality, we have decided to move it into supplementary information. Unfortunately, while above-ground disaggregation would be consistent with what we have done for the roots, this vizualisation is not possible at the moment without further model developments.*

Line 782. Equation 26 does not strike me as a core model equation, at least not for PHOREAU.
*Reply: While this equation looks similar to the generic GDD calculation used in ForCEEPS, it does tie in several of the main additions of the PHOREAU model: a tree's ability to capture light now depends on its species (through leaf phenology) and its position in the canopy (through the microclimate).*

Line 809 (Eq. 31). How is light tolerance defined?
*Reply: This was a mistake: we meant to refer to shade tolerance, a species parameter with values between 1 (shade tolerant) and 9 (shade intolerant) (L270)*

Lines 829-831. Even though this is explained in more detail in Appendix M, this paragraph comes a bit out of nowhere. Perhaps add some more context of what the bootstrapping does.
*Reply: This paragraph has slightly expanded and moved to the improvements on the ForCEEPS model section, with the other paragraphs focusing on improvements to the prediction of foliage area.*

Figure 5. Drop the word proposed? This is the implemented framework, "proposed" implies something to be developed in the future. In addition, the meaning of the arrows is a bit unclear.
*Reply: Corrected, with an expanded figure caption. Arrows indicate the relative strengths and weaknesses of each validation method.*

Lines 844-847. I do not think this is really the case. For example, Maréchaux and Chave (2017) (https://doi.org/10.1002/ecm.1271) has an extensive assessment, using multiple observation metrics.
*Reply: This statement lacked nuance and has been modified. We mostly referred to the validation of earlier generation gap-models, developed for temperate forests, which are the precursors to ForCEEPS (and so PHOREAU) like JABOWA, FORECE and Forclim, and assessed model performance on long-term predicted basal area and species distribution. (L.860-863).*

Table 1. Some rows are a bit unclear. For "Stand inventory", perhaps replace with something more specific, like "Available stand inventory information". Likewise, I was not sure about the meaning of "available soil water quantity", is this the average, maximum, minimum or the initial value? *Amended with suggestions. It is indeed the maximum (and initial) available soil water quantity.*

Line 891. The section title is "ICP II sites", but the section describes the RENECOFOR network too. *Amended.*

Line 944. Missing description for SILVAE. *A short description was included, with a fuller description in appendix T (L959-962).*

Figure 8. Is there a reason to refer to the old INRA logo instead of INRAE (which is also shown)? *None ! Amended.*

Line 986. Provide actual numbers instead of "a few dozens". *Amended, it was '40 sites'.*

Section 3.2.3 I did not follow this section. Is PHOREAU being compared with ForCEEPS, which is itself a component of PHOREAU? This struck me as rather circular....
Reply: *The comparison with ForCEEPS has been included not so much as a way to evaluate absolute model performance, but rather to evaluate the impacts (if any) of the addition of the new phenology and drought modules (a sentence was added to reflect this : L1027-1029).*

Section 4.1. Again, the sub-section title does not correspond to what is described. The first paragraph is entirely about forest structure, not water balance and plant hydraulic functioning.
*Reply: Title has been amended with the addition of "feedbacks on stand structure". During the writing of our results section, we hesitated between organizing our results by the type of variable examined (in which case the analysis of predicted ICOS stand structure should have grouped with those of ICP II simulations), or by the simulation dataset used. In the end we chose an intermediate solution, where we kept the results of simulations grouped together, but titled the sections according to the main results of interests of each simulation. In our view the prediction results for basal area and foliage area across the 4 ICOS sites are not very significant in and of themselves (at least not compared to those for the RENECOFOR and ICP II sites used later), but are needed to adequately interpret the hydraulic predictions which are the real focus of the ICOS simulations.*

Line 1055. I thought Figure W3 would help complement what is shown in Figure 9 and could be presented in the main text (likely omitting Barbeau in both figures as it does not have observations). I think this would strengthen the point that PHOREAU seems to be doing a very good job representing the basal area dynamics, though likely through compensating biases in both growth and mortality.
*Reply: The figure has been inserted in the main text. For Barbeau site, we do have observations for stand basal area, but lacking mortality data we are indeed unable to disaggregate between the effects of growth and dieback.*

Line 1073. I may have missed it, but I think Figure 12 is mentioned before Figure 11. *Indeed, figures have been interverted.*

Figure 10. The comparison for Hesse is fundamentally different from the other sites. There are many assumptions on how to scale sapflow measurements to total transpiration (which is not the same as ETR). Some context or some reference is needed for explaining how this scaling was

made, and the comparison should focus on transpiration only, not ETR.

*Reply: This was an oversight on our part. Comparison for Hesse has been restricted to predicted tree transpiration, and Figure 10, W6, and table S11 have been updated accordingly. Hourly upscaled transpiration values were directly communicated by the site PI, we do not know exactly which methods were used.*

Figure 11. Why is the colour ramp scale stretched from 0-500 cm, if all the depths in the labels are within the top 150 cm only? *Amended.*

Figure 12. Units in the Y axes should be Mpa. *Amended*

Figure 15. I think this plot is rather difficult to interpret. The comparison of basal area increments by species across multiple sites can go wrong for so many reasons, including the fact that the forest structure and composition predicted by PHOREAU does not match with the observations. I wonder if some assessment of emergent properties of the ecosystem would be more valuable in here. For example, the authors could compare how plot-scale basal area increment relates to total basal area in both PHOREAU and the observed inventories, and understand if the model has reasonable representation of canopy occupancy. Similarly, mortality rates as functions of total basal area could indicate the model ability to represent canopy thinning.

*Reply: We have addressed this point above (In the Section 3 & 4 comment reply). This validation methodology, based on the RENECOFOR exhaustive inventories, should ensure that the simulated and observed forests closely match at the start; and the short simulation length (roughly 7 years, with no intervening cuts) should avoid major drifting. We do agree that tree-wise validation is a less meaningful test than stand-wise comparison for this kind of model; this had been discussed in the text, but we have now moved the tree-wise plot to appendices to make this clearer.*

Section 4.5 and Figure 17. How are the "accurate prediction", "partially accurate prediction" and "false prediction" are quantitatively defined? Without knowing these thresholds, I think there is little value showing this figure. In addition, panel (a) overlays too many colours, and it is rather difficult to identify the patterns of model performance. Perhaps the authors could use a white background in panel (a), and the same colour code and same symbols in both panels to make these figures a bit more comparable.

*Reply: The criteria for classifying a prediction as accurate, partially accurate or failed were defined in the panel caption; for further clarity we have also included the definition in section 3.2.4 (L1061-1065). Regarding the background niche colors, while we aknowledge it can make the interpretation of the first graph more difficult (this was actually discussed at length between co-authors), we feel it is necessary to link the geographical representation with the niche-based*

*one, by showing to which niche each simulation point belongs. This has been detailed in the caption for figure b.*

Lines 1254-1256. I think similar assessments exist for other models, if not individually based, at least cohort-based models (which seems to be the approach used by PHOREAU in any case). For example Xu et al. (2021) (https://doi.org/10.1111/nph.17254) and Eller et al. (2020) (https://doi.org/10.1111/nph.16419) have to some extent assessed similar characteristics.
*Reply: We have amended and restricted this too broad statement to gap models, and in particular individual-based gap models (L1213-1219). While we have used, in this validation, the option to aggregate tree hydraulics across tree size classes, the overall functioning of the model is still largely individual-based. PHOREAU is already being used without this option on a fully individual basis (Louis Devresse in prep.)*

Lines 1271-1277. In principle, I agree with this discussion point, but there is a downside in the simplification too. Models that are too simplistic may lack mechanisms to represent how forests would respond to shifts ti no-analog conditions, such as changing climate or changing disturbance regimes.
*Reply: This paragraph was indeed too one-sided and has been amended (L1240-1243): PHOREAU aims to strike a balance between fully ecophysiological models and more simplified gap models, by focusing on the physiological mecanisms most likely to be affected by changing climatic conditions, while maintaining a more simplified representation for growth which in turn allows the model to more accurately predict (at least for present conditions) overall stand structures. Without baseline realistic predictions of stand foliage and basal areas, our assessment of the effects of drought on forest functioning would risk being flawed from the outset.*

Lines 1333-1342. This was not extensively assessed in this manuscript, so I think before implementing more complex approaches, it may be useful to test whether these new processes are indeed needed, or if the results as they are are already reasonable.
Reply:We agree with the reviewer about the relevance of a parsimonious approach when implementing new processes in a model (this as actually discussed in ForCEEPS seminal paper Morin et al. 2021 – cited in the manuscript*). In this manuscript, we have integratively assessed the effects of the integration of phenology on stand growth and tree water stress. The phenology results themselves were not assessed directly, because they are the direct, unmodified results of the PHENOFIT model which has been extensively tested and validated in other publications. However, while this simplified our validation protocol, we feel it is ultimately disappointing to not take advantage of the available forest structure data to inform phenology results. We discuss here this possibility (L1307-1309), which, while promising, would require a*

*validation on scarce forest inventory data with both microclimatic and phenological measurements.*

Line 1362-1371. Use HEH instead of EHE? *Amended*.

---

## Author Response (AR2)

We sincerely thank the reviewers, the topic editor, and the editorial support team for the time and effort dedicated to evaluating this extensive manuscript.

In the revised version, we have addressed all remaining points raised. Specifically, we increased the figure font sizes, corrected the typo in the MPa unit, modified the caption of Figure 4, and added further details on the handling of patches in PHOREAU.

Regarding the cube root in Equation (3), we note that this formulation originates from the original ForCEEPS presentation and validation study. The model was calibrated and validated using this approach, which does indeed penalizes trees with strong reductors more than would be the case if a ½ power formulation were used.